# Understanding the mechanism of facilitation in hoverfly TSDNs

**Anindya Ghosh** [1]*, **Sarah Nicholas**[2], **Karin Nordström**[2], **Thomas Nowotny**[1], **James Knight**[1]

1 Sussex AI, School of Engineering and Informatics, University of Sussex, Brighton, United Kingdom,
2 Flinders Health and Medical Research Institute, Flinders University, Adelaide, South Australia, Australia

* anindya128@yahoo.com

**Data availability statement:** Electrophysiology data have been deposited on Figshare https://doi.org/10.25377/sussex.28615313. The

## Abstract

Many animals use visual motion cues to track and pursue small, fast-moving targets, such as prey or conspecifics. In target-pursuing insects, including dragonflies and hoverflies, Small Target Motion Detector (STMD) neurons are found in the optic lobes and are believed to be presynaptic to Target Selective Descending Neurons (TSDNs) that project to motor command centres. While STMDs respond robustly to target motion – even when displayed against moving backgrounds – TSDN target responses are modulated by background motion. Depending on whether the background motion is syn- or contra-directional to the target motion, the response of the TSDNs is either suppressed or facilitated (amplified). This suggests that TSDNs not only receive input from STMDs but also from neurons sensitive to background motion, but this neural circuit is not clearly understood. To explore the underlying neural mechanisms, we developed three candidate TSDN circuit models – which combine input from bio-plausible STMDs and optic flow-sensitive Lobula Plate Tangential Cells (LPTCs) in different ways – and fitted them to published electrophysiology data from hoverfly TSDNs. We then tested the best-fitting models against new electrophysiological data using different background patterns. We found that the overall best model suggests simple inhibition from LPTCs with the same preferred direction as the STMDs feeding into the TSDN. This parsimonious mechanism can explain the facilitation and suppression of TSDN responses to small targets, and may inform similar studies in other animals.

## Author summary

Many human sports, including tennis, football, and basketball, rely on the ability to visually detect and respond to the motion of a small, rapidly moving object. Indeed, some sports stars seem to have an uncanny ability to predict the future location of a ball. Similarly, many animals also need to detect the motion of small objects, as these may represent an approaching predator, prey, or a conspecific that needs to be avoided or

code is available on GitHub https://github.com/anindyaghosh/TSDN_facilitation.

**Funding:** AG was funded by the Leverhulme Trust, KN by the US Air Force Office of Scientific Research (AFOSR, FA9550-23-1-0473) and the Australian Research Council (ARC, DP210100740 and DP230100006), TN by the EPSRC (grant EP/S030964/1) and the EU (grant no. 945539), and JK was funded by the EPSRC (grants EP/V052241/1 and EP/S030964/1). The funders had no role in study design, data collection and analysis, decision to publish, or preparation of the manuscript.

**Competing interests:** The authors have declared that no competing interests exist.

pursued. Insects, in particular, which have small brains with very modest power requirements, appear to solve the problem of visually detecting small targets effortlessly. Male hoverflies naturally engage in complex pursuit behaviour, and since they have specialised neurons for target tracking, hoverflies are ideal for understanding the underlying neural circuitry. Here, we combine neural recordings from hoverflies with computational modelling to show how neurons that respond selectively to target motion are combined with neurons that respond to widefield optic-flow to generate behaviourally relevant sensorimotor responses. Importantly, our model does not involve learning and hence suggests that these behaviours can be innate.

## Introduction

In many species, the ability to detect small, moving targets in visual clutter is crucial for survival. For example, for a predator, these targets may represent prey, while in other animals, they may represent rivals or potential mates [1]. Thus, neural circuits underlying target-detection have been heavily studied and target-detection behaviours have been described in a range of species, including salamanders [2], zebrafish larvae [3], dragonflies [4], and hoverflies [5].

In insects, target-detection behaviours are believed to be driven by target-tuned neurons found in the visual and pre-motor regions [6]. Specifically, in hoverflies, which pursue conspecifics and other territorial intruders at high speed, there are target-tuned neurons in the optic lobes – the Small Target Motion Detectors (STMDs) [7,8] – as well as in the cervical connective – the Target Selective Descending Neurons (TSDNs) [9]. These descending neurons connect the central brain with motor command centres in the thoracic ganglia. STMDs are able to respond robustly to small, moving targets in visually cluttered environments, even when there is no relative motion between the target and the background [10]. In contrast, the responses of their presumed post-synaptic targets, the TSDNs, are modulated by background motion [11]. Indeed, while TSDNs respond strongly to targets moving across either homogeneous or cluttered stationary backgrounds, if the background pattern moves in the same direction as the target, the TSDN response is comparatively suppressed. Conversely, if the background moves in the opposite direction, the TSDN response is facilitated [11].

The outwardly surprising differences in the responses of STMDs and the TSDNS – which we assume are postsynaptic to STMDs – may be explained by hoverfly behaviour. Hoverflies perform two-stage target pursuit: (1) they start with an open-loop interception response [12], and (2) once close to the target, they transition to a 'shadowing' or following behaviour, matching their speed to that of the target to maintain a close distance [5]. Throughout flight, they perform body saccades [13], which generate widefield optic flow. This motion is detected by lobula plate tangential cells (LPTCs), which in turn project to optic flow-sensitive descending neurons that mediate the resulting optomotor response [14–16]. Thus, the robust target response provided by STMDs [7,8] can alert the central brain of the motion of a target irrespective of background motion, while the LPTCs inform the animal about self-motion. We hypothesise that if the TSDNs combine the input from STMDs and LPTCs, they can project the information needed for flight correction. Specifically, when the target and background move in the same direction, the optomotor response itself ensures that the insect follows the target, and no extra TSDN signal is needed. In contrast, if the background moves in the opposite direction to the target, the TSDNs need to overcome the optomotor response, and therefore, the TSDN signal needs to be increased (for discussion, see [11]).

There are many models for the detection of motion (for review, see e.g. [17]), with interesting analogies between insects and vertebrates [18]. One of the classic phenomenological models, the Hassenstein-Reichardt Elementary Motion Detector (EMD) model performs a non-linear comparison of signals from two points in space after delaying one of them [17,19–24]. More recent biophysical models based on T4 and T5 cells in *Drosophila*, rely on the integration of fast excitatory signals with spatially offset slow inhibitory signals [24,25]. While all these models generate directional responses to widefield motion, some may also respond to smaller objects (as LPTCs do [26]). However, even if LPTCs respond to smaller objects, they are not target-tuned. Therefore, other models have been developed to explain the mechanisms underlying the ability of STMDs to detect small targets amidst clutter [7,27]. These generally either use a centre-surround mechanism [28] or cascade the centre-surround mechanism with temporal correlation filters [29,30]. Temporal correlation filters, often referred to as 1-point correlators, are based on the comparison of ON (brightness increment) and OFF (brightness decrement) signals from one point in space after delaying one of them. Thus, an elementary STMD (ESTMD) [29], which implements a 1-point correlator, detects the unique temporal signature associated with a dark target – where a leading OFF edge is immediately followed by a trailing ON edge – and thus only responds to small, dark, moving objects while being generally unaffected by background motion. A consequence of the 1-point correlator model is that, if target-like spatio-temporal signatures are found in the background, they will also elicit a response. This is indeed seen in dragonfly STMDs [30]. In addition, both STMDs and TSDNs respond poorly to signals containing only leading OFF or trailing ON edges [11,30], suggesting similar ESTMD-type input, and supporting the notion that TSDNs are post-synaptic to STMDs.

Existing models of small-object detection [28,29,31] fail to explain the observation that hoverfly TSDNs *are* affected by background motion [11]. Therefore, additional mechanisms are needed, and we hypothesised that both STMDs and LPTCs are necessary to describe the TSDN circuit mechanistically. While there are published models that incorporate ESTMD-EMD interactions, these either act to make the model ESTMD output directionally-selective [32] or to *enhance* target detection against velocity matched, syn-directional background motion [33]. As the responses of biological TSDNs are suppressed under these conditions, these models fail to explain the physiology.

To test our hypothesis, we made the outputs of the ESTMD model [29] directionally-selective [32], tiled them over the visual field, and spatially pooled their outputs to match biological STMD [34] and TSDN receptive fields [11]. We modelled optic-flow-sensitive LPTCs using EMDs (Hassenstein-Reichardt detectors) [20] and then fitted plausible TSDN circuit arrangements, combining the STMD and LPTC outputs, to previously published electrophysiological data from TSDNs [11]. Subsequently, we acquired new biological data where TSDNs responded to targets against different backgrounds and tested the candidate circuits on this new data to assess whether the models generalise. This led us to discard one of the circuits. To determine the most viable of the remaining two candidate circuits, we then tested them on other stimuli described in [11], allowing us to eliminate another circuit. Finally, we discuss how this simple circuit can mechanistically explain observations from biological TSDNs. Thus, by combining existing neuron models of STMDs and LPTCs, we obtain a circuit configuration that can, for the first time, plausibly explain the phenomenon of facilitation and suppression observed in biological TSDNs [11].

## Results

To model the responses of hoverfly TSDNs to targets moving against background motion, we took advantage of existing ESTMD [29,35] and EMD models [17,19]. To facilitate the continuity of the ESTMD model literature, we use the same naming conventions for the mechanisms described here as prior work [29,35].

### LPTC model captures direction and velocity tuning of biological LPTCs

LPTCs have been studied extensively in a range of dipteran flies, including the blowfly *Calliphora*, the fruitfly *Drosophila*, and the hoverfly *Eristalis*. There are 45-60 LPTCs in each hemisphere [36], including neurons that give graded responses, spiking responses, and a mix [37]. Here, we developed models based on spiking neurons, and as *Callipora* H1 and H2 are the best-studied spiking LPTCs [38–41], we used published data for these neurons for model fitting. However, considering that the velocity tuning, contrast response functions, and direction sensitivity are strikingly similar between diverse species [42–44], this choice should not affect our results substantially.

We modelled LPTCs by spatially pooling output from local elementary motion detectors (EMDs) (see Lobula Plate Tangential Cell (LPTC) modelling section) and compared our model output to biological data from blowfly H2 neurons (Fig 1). Using a 7.4° diameter dark dot stimulus on a white background (Michelson contrast of 60%), Longden et al. [45] obtained the local motion direction tuning of the H2 neuron at 0° elevation and 30° azimuth. We used a similar dot stimulus to obtain local motion direction tuning for our model LPTC (red data, Fig 1A), with a preferred direction to the right (0°) While there are other models for LPTC input [24], our EMD-based model (Fig 1) also closely matches the temporal frequency tuning of the biological LPTC in both the preferred (green arrow) and anti-preferred direction (purple arrow). This was tested using a vertical square-wave grating with a wavelength of 11.4° and Michelson contrast of 95%, presented for 0.5 s. In both biology and the

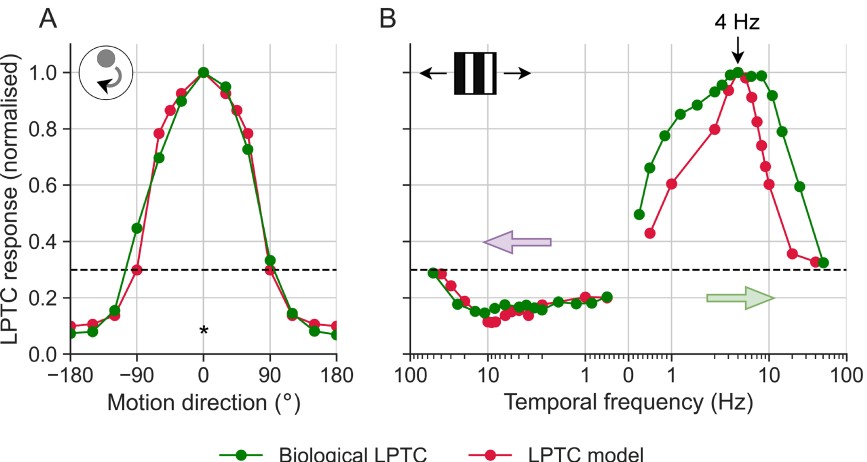

**Fig 1. Velocity tuning of model and biological LPTCs.** (**A**) Local motion direction tuning curve obtained using a dot stimulus (see ref [45] for stimulus details). The asterisk (*) denotes the preferred direction. (**B**) Temporal frequency tuning to an 11.4° wavelength, square-wave vertical grating stimulus, with Michelson contrast of 95%. The green and purple arrows indicate motion in the preferred direction (PD) and null direction (ND), respectively. In both panels, the dashed black line denotes the spontaneous activity, and the model data represents the mean normalised response over 0.5 s stimulation. The biological LPTC data was obtained from ref [45].

model, the maximal response occurs around 4 Hz. The model output shows a narrower temporal frequency tuning than the biological data in the preferred direction (Fig 1B), which could be changed by adding a saturating non-linearity to the model output. However, the biological and model tuning curves both have their peak normalised response at the same temporal frequency, which is close to the velocities relevant to our work. In addition, both Fig 1A, 1B show that our model exhibits direction opponency, with an asymmetry about the spontaneous rate in the response amplitude to motion in the preferred and anti-preferred directions (see e.g. [42]).

Longden et al. [45] found that biological LPTCs start responding to square-wave grating stimuli 20–60 ms after stimulus onset (green data, Fig 2A). In response to a 4 Hz stimulus, our model captures the response onset dynamics of biological LPTCs well (red, Fig 2A). We quantified the mean response within a 20–60 ms time window across temporal frequencies and found that our model LPTC mimicked biological data (Fig 2B). Compared to the tuning curves in Fig 1B, those in Fig 2B peak at a higher temporal frequency. This can be attributed to the fact that LPTCs respond faster as the temporal frequency increases [46], and that the velocity response functions in Fig 2A and 2B are obtained through the mean LPTC responses over 0.5 s and 20–60 ms time windows respectively. In summary, our LPTC model responses closely match those of their biological counterparts (Fig 1 and Fig 2).

## The dynamics of our STMD model match hoverfly biology

STMDs respond robustly to targets moving against background motion, even without relative motion between the two [10]. Wiederman et al. [29] and Bagheri et al. [35] proposed an elementary STMD (ESTMD) model that encapsulates this behaviour, which we reproduced (see STMD modelling for algorithmic details of the STMD model). In this model, each delayed OFF signal is correlated with an undelayed ON signal from the same point in space. Together with spatial centre-surround inhibition, this creates a selective detector for a dark target moving on a brighter background. To better match hoverfly response functions,

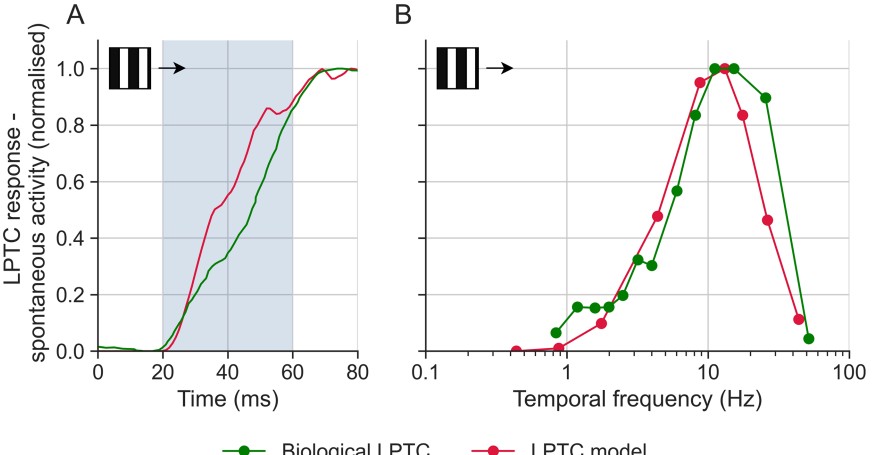

**Fig 2. Response onset characteristics of model and biological LPTCs.** (**A**) The response onset to a 4 Hz stimulus, smoothed with a Gaussian filter with $\sigma$ = 6 ms. The stimulus appeared and started moving at 0 ms. The shaded region shows the analysis window used in panel B. (**B**) The transient response (averaged over the 20–60 ms period shown in panel A) to preferred direction motion as a function of temporal frequency. The spontaneous activity has been subtracted. The biological LPTC data was obtained from ref [45]. All responses are normalised to their peak.

we modified the Gaussian kernel used for spatial blurring (see Early visual processing in S1 Appendix) and the temporal adaptation time constants (see Target matched filtering in S1 Appendix), but kept most other components similar.

Hoverfly STMDs vary in their directional selectivity as well as receptive field size and location, with up to 20 different physiological types identified [10]. All STMDs are defined by their sharp size tuning, which can be determined using a black bar with a fixed width (parallel to the direction of target motion) and varying height (perpendicular to the direction of target motion) moving at a constant speed. Hoverfly STMDs give peak responses to targets with a height of a few degrees of the visual field (green data replotted from [34], Fig 3A). Similar size tuning is found in other STMDs [8]. Our ESTMD model closely matches Wiederman et al.'s [29] model (compare black and red data, Fig 3A), but is narrower than biology (green, Fig 3A). However, at the relevant size (∼1.4°) used in this paper, there is a close match between biology and models.

We next look at the velocity response functions. As there is no published data from male STMDs, we used data from females to investigate the model's ability to recapitulate the velocity response function (green data replotted from [7], Fig 3B). The female STMD velocity sensitivity was determined using a black square of fixed size moving at a range of speeds in the neuron's preferred direction. Fig 3B shows the resulting velocity response function, which in models and experiments has its peak response around 30–200°/s [7,29]. While the biological velocity tuning is broader than the models, Fig 3B shows that our ESTMD model matches Wiederman et al. [29]'s ESTMD model well, and is similar to biology at the relevant velocity (∼90°/s).

As TSDNs are directionally selective [1], they likely receive input from directionally selective STMDs. To capture this in our modelling, we passed the output from neighbouring ESTMDs through an EMD, resulting in directional selectivity (similar to [32]). In addition, we spatially pooled output from the directionally selective ESTMDs to match the receptive fields of small-field STMDs [1,34]. Details about the directionally-selective STMD model are in the STMD modelling section in the Methods.

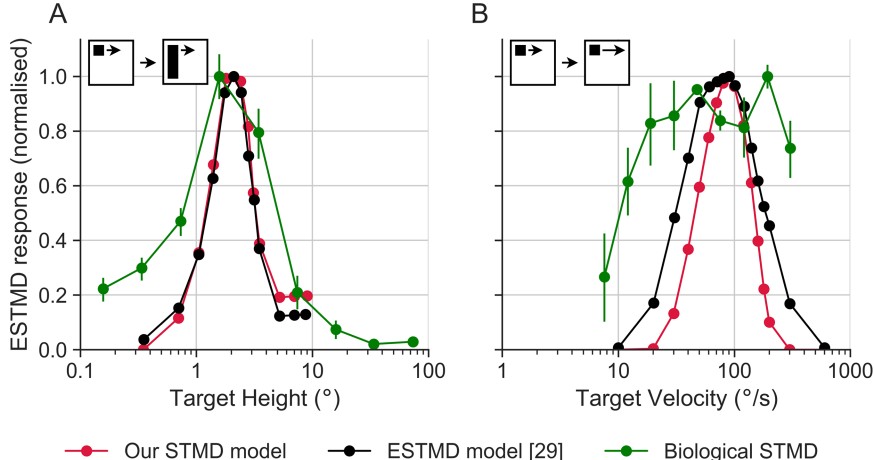

**Fig 3. Comparison of spatiotemporal characteristics from model and biological STMDs.** (**A**) Normalised responses to targets of different heights. The bar had a constant width of 0.8° and was moving at 50°/s. In modelling, we used bar heights between 0.35° and 9°. The biological STMD data is from ref [34] (**B**) Velocity tuning of black, square targets (0.8° × 0.8°). In modelling, we used velocities in the 10–600°/s range. The biological STMD data is from [10]. The black data is from [29]. Please note that the targets and bars in the pictograms are not to scale.

## TSDN circuit arrangements

TSDNs are likely to be post-synaptic to STMDs [6,47], as supported by their size, contrast, and velocity response functions [1], as well as their response to ON and OFF edges [11]. TSDNs are also highly directional with a preference for horizontal motion away from the visual midline [1]. Unlike LPTCs, TSDNs do not respond to motion in the anti-preferred direction and do not have a spontaneous firing rate. Contrary to dragonfly TSDNs, which constitute a heterogeneous population [48], hoverfly TSDNs are largely homogeneous, differing mainly in their directional selectivity [1]. An important difference compared to STMDs is that TSDN responses *are* affected by background motion, suggesting that they receive additional input from LPTCs. Syn-directional background motion inhibits TSDN response to target motion, where the level of inhibition is proportional to the level of LPTC excitation generated by the particular background [11,47]. Therefore, as the strength of TSDN inhibition follows the direction, velocity, and contrast tuning of optic-flow sensitive neurons [11,47], it seems plausible that they receive inhibitory input from LPTCs with the same preferred direction. In addition, TSDNs are facilitated by contra-directional background motion [11], suggesting that they receive input from neurons with the opposite preferred direction.

We based our modelling on published data from hoverfly TSDNs responding to small circular targets moving over a 'starfield' background [11]. This starfield pattern covers the entire visual field and was designed to simulate the type of optic flow generated by self-motion through the world, including e.g., sideslip translations and yaw rotations [14]. Because this background covers a large part of the visual field, it will strongly stimulate optic-flow sensitive neurons [14] and, because it consists of many target-sized features moving coherently (Fig 1 of [11]), it *could* also stimulate STMDs, although this has yet to be shown in biology. However, direct additive inputs from LPTCs or STMDs can be discarded categorically for the following reasons:

1. It is not plausible that LPTCs directly excite the TSDN, as this would result in TSDN responses to wide-field optic flow even in the absence of a target. This does not occur in biological TSDNs [11,47].
2. It is not plausible that STMDs with a directional preference opposite to that of the TSDN directly excite the TSDN, as this would result in TSDN responses to both directions of motion, which does not occur in biological TSDNs [1].

Thus, we evaluate the three simple, bio-plausible circuit arrangements illustrated in Fig 4):

**Circuit 1** involves a linear summation of STMD and LPTC inputs.

**Circuit 2** is an extension of circuit 1, with additional excitatory input from an LPTC with the opposite preferred direction. This LPTC enhances any STMD signals, but does nothing if the STMD is inactive. Thus, in many cases, the outputs of circuits 1 and 2 are the same. However, in the event of background motion contra-directional to target motion, the STMD signal would receive an additional *boost* in circuit 2, thus resulting in a facilitated TSDN response.

**Circuit 3** also amplifies the STMD input to the TSDN. However, in circuit 3, the amplification comes from an STMD with the opposite preferred direction of motion. For facilitation, circuit 3 thus relies on the background consisting of target-like objects, such as the 'starfield' background used in the published recordings of biological TSDNs [11].

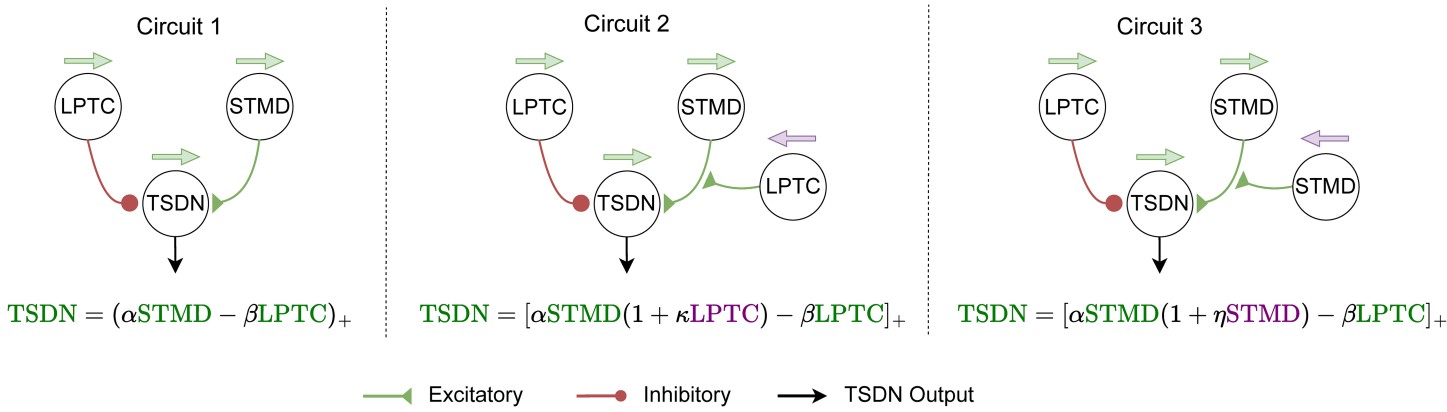

$$\text{TSDN} = (\alpha\text{STMD} - \beta\text{LPTC})_+ \qquad \text{TSDN} = [\alpha\text{STMD}(1 + \kappa\text{LPTC}) - \beta\text{LPTC}]_+ \qquad \text{TSDN} = [\alpha\text{STMD}(1 + \eta\text{STMD}) - \beta\text{LPTC}]_+$$

◁── Excitatory     ●── Inhibitory     ⟶ TSDN Output

**Fig 4. Candidate Model Circuits**. All TSDN circuits include direct excitatory input from STMDs. *Left*. Circuit 1 involves a simple linear subtraction of the LPTC response from the STMD response. All neurons in the circuit have the same preferred direction (green arrows). *Middle*. Circuit 2 additionally includes an LPTC with the opposite preferred direction (purple arrow), multiplicatively amplifying the STMD signal. *Right*. Circuit 3 involves the addition of an STMD with the opposite preferred direction (purple arrow), multiplicatively amplifying the signal from the STMD. The circuits are reflected in the functional equations below each. $\alpha$, $\beta$, $\kappa$ and $\eta$ are the coefficients of the relevant terms.

In all three circuits, if a hoverfly is stationary or *hovering*, due to a lack of self-generated optic flow, the only input from the LPTC would be its spontaneous rate, and the TSDN output would be similar to the output of a TSDN receiving only STMD input.

## Comparisons of candidate TSDN circuits

We fit the neuronal weight coefficients ($\alpha$, $\beta$, $\kappa$ and $\eta$ from Fig 4) and the LPTC equation (Lobula Plate Tangential Cell (LPTC) modelling section) coefficients ($g$, $\gamma$ and $s$) of our three candidate TSDN circuits to minimise the average root-mean-square error (RMSE) of model output against spike density functions obtained from published spike train data of several TSDNs responding to targets moving on a starfield background (see Data analysis and statistics section in Methods) [11]. In these experiments, the background consisted of thousands of targets moving horizontally to mimic the perspective-corrected optic flow generated by the viewer sideslipping at 50 cm/s [14]. In circuit 2, where two LPTCs are present, we only optimised a single set of LPTC coefficients. We use an 'STMD only' circuit as a benchmark to normalise the circuit errors. The 'STMD only' circuit comprises STMD output weighted by the coefficient $\alpha$. The coefficient values for the best fit are shown in Table 1.

Fig 5 shows the biological (green data replotted from [11]) and model responses of a single TSDN to a target moving over either a bright background or over a starfield background. When the target moves across a bright screen or against a stationary pattern ('alone' and 'stationary', Fig 5), all the outputs are similar. However, when the target is displayed over a

**Table 1**. Neuronal weight and LPTC equation coefficients.

| Circuit | LPTC equation | | | Neuronal weight | | | |
|---|---|---|---|---|---|---|---|
| | $g$ | $\gamma$ | $s$ | $\alpha$ | $\beta$ | $\kappa$ | $\eta$ |
| STMD only | - | - | - | 0.4 | - | - | - |
| Circuit 1 | 13 | 0.3 | 10 | 0.8 | 13.9 | - | - |
| Circuit 2 | 11 | 0.9 | 10 | 0.3 | 29.9 | 2 | - |
| Circuit 3 | 13 | 0.3 | 10 | 0.5 | 4.7 | - | 1 |

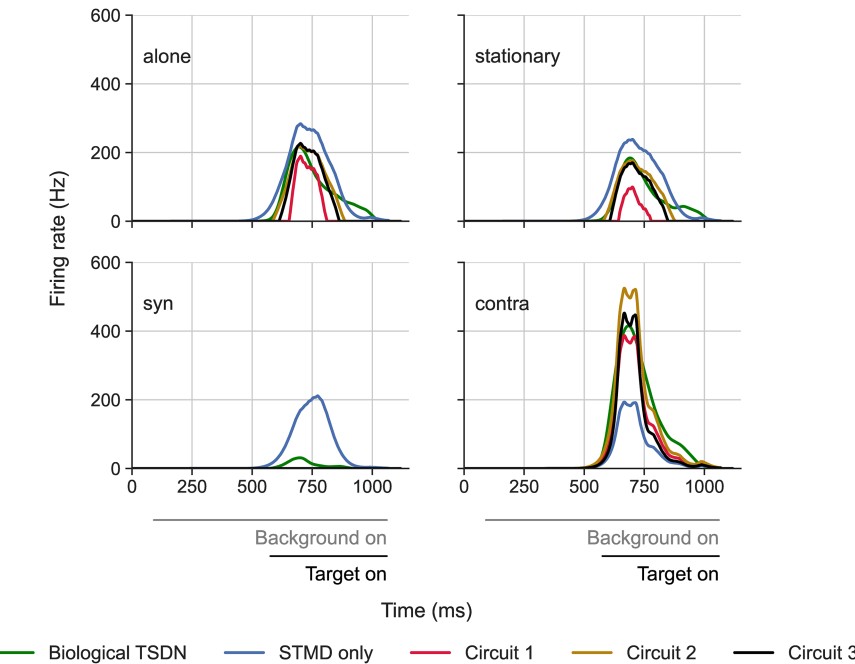

**Fig 5. Example TSDN model and biological outputs to targets moving over the starfield background.** The data show the spike histograms as a function of time from a single TSDN as it responds to a small target moving through its receptive field, in the neuron's preferred direction, at 900 pixels/s. The biological data (green) is replotted from [11]. The 'STMD only' circuit (blue) represents a TSDN circuit consisting only of an STMD input (no LPTC input). The different panels show the response to the black, circular target moving over a bright screen ('alone'), over a stationary starfield pattern ('stationary'), a sideslipping starfield (50 cm/s in the same direction as the target ('syn'), or in the opposite direction ('contra')). The greyscale bars under the data indicate the presence of the stimuli.

starfield background moving in the same direction as the target, all TSDN circuits are strongly inhibited ('syn', Fig 5). In contrast, the modelled 'STMD only' output remains strong, as expected from biological findings that STMDs are unaffected by background motion (blue data in Fig 5; [10]). When the target is displayed over a starfield background moving in the opposite direction, all TSDN circuits are facilitated ('contra', Fig 5), whereas the modelled 'STMD only' output remains largely unaffected. Again, this is expected from biology where STMDs are unaffected by background motion (blue data in Fig 5; [10]). Note that the 'contra' response from circuit 2 seems to 'overshoot' the biological TSDN response.

LPTCs have a spontaneous rate, i.e., they fire at a non-zero baseline rate even when not stimulated [45] (dashed lines, Fig 1). When stimulated with motion in the preferred direction, the response increases, whereas motion in the non-preferred direction leads to an inhibition of the firing rate (see e.g. Fig 1B). The LPTC spontaneous rate means that all the proposed circuits receive inhibitory LPTC input even in the 'alone' and 'stationary' conditions. However, for circuit 3, since in 'alone' and 'stationary' conditions, only the target moves and STMDs do not have a spontaneous rate, the output of the STMD with the opposite preferred direction (purple arrow, Fig 4) is zero. This allows the synaptic weight of this amplifying STMD, $\eta$, to be tuned specifically towards minimising the fitting error for the 'contra'-directional condition. Indeed, for this example neuron (Fig 5), circuit 3 fits the biological TSDN data remarkably well.

To compare this across neurons, we calculated the average RMSE for each circuit per condition (Fig 6) normalised against the equivalent RMSE of a TSDN circuit with only an STMD

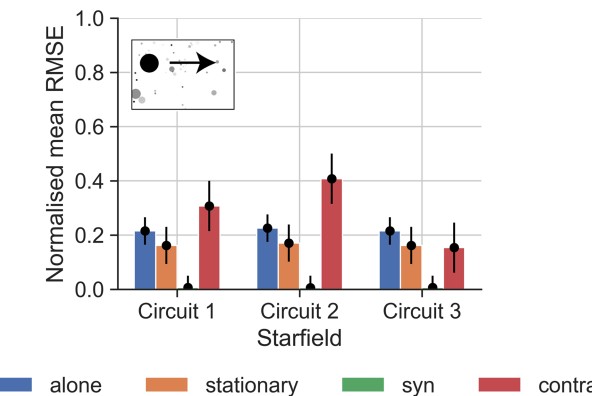

**Fig 6. Errors from fitting candidate circuits to biological TSDN data for targets moving over the starfield background.** For each neuron (N=35, data from [11]), we calculated the mean RMSE by comparing each circuit (described in Fig 4) with the electrophysiological data. The mean RMSE of each circuit per condition was then normalised by the equivalent RMSE of the 'STMD only' circuit. The whiskers of the black points (the mean RMSEs) indicate the variation across neurons, quantified as the standard deviation of the normalised mean RMSE of the particular bar plot. Note that the background in the pictogram has been faded out and that the target has been magnified substantially for viewing purposes.

input ('STMD only'). For each neuron simulation, we used the same initial conditions, i.e., the same starting points for the target as were used in the electrophysiology experiments, and took each neuron's receptive field into account (see TSDN receptive fields for details). We found that 'syn' resulted in the smallest errors for all circuits, whereas 'contra' led to the largest errors. However, it should be noted that with syn-directional motion, TSDN responses tend to be suppressed to near-zero values (Fig 6; [11]). This is because the 'STMD only' circuit, which we used for normalising, lacks an LPTC component and thus, inherently attempts to *average* across the hyperparameter search space. This can be seen in the 'syn' panel of Fig 5 where the 'STMD only' response shows no suppression.

Across background conditions, and across neurons, circuit 1 performs better than circuit 2, and circuit 3 outperforms both circuits 1 and 2 (Fig 6). All circuits outperform the 'STMD only' circuit – the normalised errors are all below 1. Thus, across neurons, the normalised average RMSEs reflect the quality of the circuit fits exemplified for one neuron in Fig 5.

## TSDN responses to targets moving across broadband backgrounds

When fitting the different circuits (Fig 6), we used published biological data where TSDNs responded to targets moving over a starfield background [11]. As the starfield background used in those experiments consists of thousands of target-like features, it *could* stimulate STMDs [27,29] (although this remains to be shown experimentally). As this makes it difficult to separate the different circuit configurations (Fig 4), we generated new electrophysiological data (see Electrophysiology) where TSDNs responded to targets moving against two other backgrounds – 'cloud' (N=10) and 'sinusoidal' (N=6) – which do not stimulate STMDs [10].

The cloud background has a 2D power spectrum similar to that of typical natural scenes and a mean luminance of 50% [47] with a Michelson contrast of 100% (see ref [7] and Visual stimuli for more details). Consistent with earlier work [47], TSDN responses to target motion are inhibited when the target is displayed against syn-directional 'cloud' background motion (Fig 7C, 7F, 7I). Earlier work found that a similar contra-directional, naturalistic background

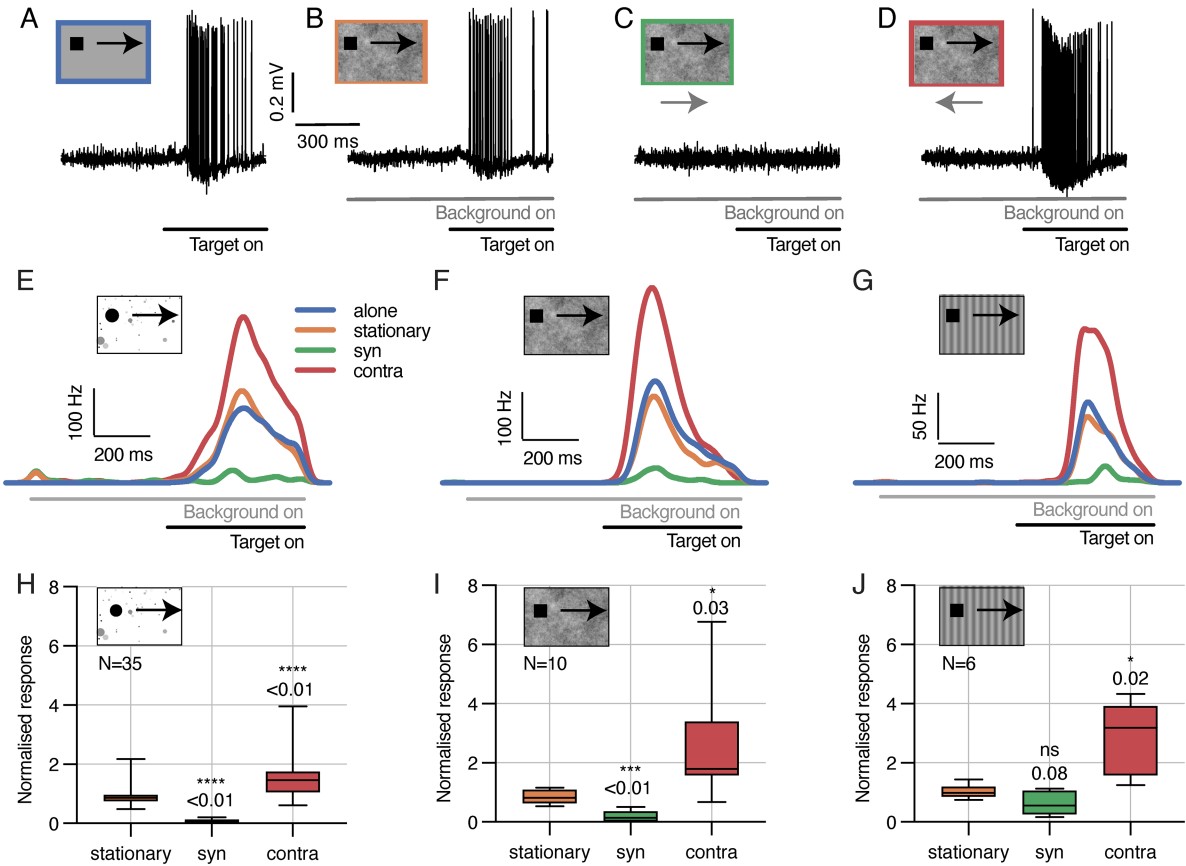

**Fig 7. New electrophysiological recordings of TSDNs responding to targets moving over cloud and sinusoidal backgrounds. (A-D)** Example raw electrophysiology data from a single TSDN responding to targets moving over a grey background ('alone'), or a 'stationary', 'syn'-directional, or 'contra'-directional cloud background. The greyscale bars under the data show the stimulus timing. **(E-G)** Spike rate responses from three different TSDN neurons for four background conditions, using the starfield (E), cloud (F), or sinusoidal grating (G) background. In all cases, the data show the spike histograms averaged across repetitions (starfield: n=9-15 repetitions; cloud: n=24 repetitions; sinusoidal: n=30 repetitions), and converted to rates as described in Methods. **(H-J)** Boxplots of TSDN responses across neurons for starfield, cloud and sinusoidal backgrounds. The whiskers of the boxplots extend across the range of the data (min to max). In all cases, we calculated the average spike frequency across the entire time the target was on the screen and normalised this to the response to a target moving over a bright (panel H), or grey (panels I and J), background. The data in panels E, H are replotted from [11]. Stars indicate statistical significance using one-way ANOVA followed by Dunnett's multiple comparisons test. Note that the pictogram targets have been magnified substantially for viewing purposes.

inhibited TSDN responses to target motion [47]. However, here we find that they are strongly facilitated by contra-directional cloud background motion (Fig 7D, 7F, 7I).

The 'sinusoidal' background is a vertical sinusoidal grating with a wavelength of 7°, moving at 5 Hz, and a Michelson contrast of 25%, which should not stimulate any STMDs [10]. We found that this background also inhibited TSDN responses when targets moved syn-directional, and facilitated TSDN responses when contra-directional (Fig 7G, 7J).

The new backgrounds not only allow us to test the robustness of all the fitted circuits, but also let us test the viability of circuit 3 as it relies on STMD input for the facilitation, and thus on backgrounds (like 'starfield') that contain target-like features. Because both the cloud and sinusoidal backgrounds are devoid of target-like features, the facilitation seen in the new electrophysiology (Fig 7) argues strongly against circuit 3.

Using the circuit coefficients listed in Table 1 (which we *fitted* on the starfield dataset), we *tested* each candidate circuit on the cloud and sinusoidal datasets (Fig 7). Similar to our initial testing on the starfield dataset, we used the same initial conditions, i.e., same starting points of the target, as in electrophysiology, and took each neuron's receptive field (see TSDN receptive fields for details) into account in the simulations. Again, the errors between the biological and modelled TSDNs were quantified by an average RMSE per neuron.

We found that the circuit 1 errors for the cloud and sinusoidal backgrounds were fairly consistent with those for the starfield background (compare Fig 6 and Fig 8). However, circuits 2 and 3 produced much higher errors (Fig 8) compared to those obtained on the starfield background, especially for contra-directional background motion. For circuit 2, this might be due to the model output for contra-directional background motion being significantly higher than the biological TSDN response (see single neuron response and modelling in Fig 5), which resulted in higher errors across neurons (Fig 8). For circuit 3, the large errors stemmed from the lack of target-like objects in the background, and the circuit's subsequent failure to produce facilitation. All three circuits had higher errors to syn-directional sinusoidal background motion (Fig 8) compared with the starfield background (Fig 6). This is because some TSDNs responded to targets moving across the 'syn'-directional sinusoidal background (Fig 7J), resulting in model testing errors. Nevertheless, taken together (Figs 7 and 8), these results suggest that circuit 3 can be discounted as a viable TSDN circuit.

## Circuit 1 provides best fit to results of preceding optic flow experiments

Nicholas and Nordström [11] explored whether the TSDN response would also be affected by background motion that immediately preceded, but did not coincide with target motion (Fig S3 of [11]). The stimuli consisted of two parts: (1) preceding optic flow – background motion *before* the moving target appears, and (2) concurrent optic flow – background motion *while* the target is in motion.

The effects of preceding optic flow on TSDN responses are thought to relate to the overlap of decaying LPTC responses and rising STMD activity and hence depend strongly on the

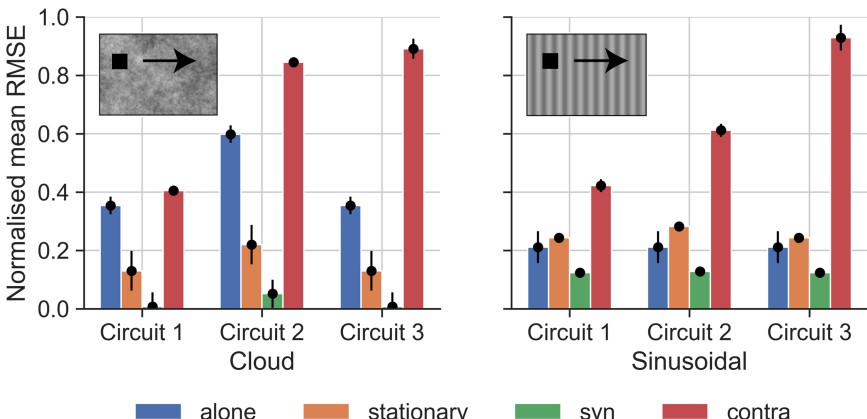

**Fig 8. Errors from testing candidate circuits on biological TSDN data for cloud and sinusoidal backgrounds.** The normalised mean RMSEs from fitting each circuit described in Fig 4 to TSDN electrophysiological data for each condition for the cloud (left) and sinusoidal (right) backgrounds. The RMSEs of each circuit per condition are normalised by the equivalent RMSE of the 'STMD only' circuit. The whiskers of the black points indicate the standard deviation of the normalised mean RMSE of the particular bar plot. Note that the targets in the pictograms have been magnified substantially for viewing purposes.

offset and onset latencies. Therefore, to investigate how the preceding optic flow affects our
model outputs, we first investigated the response latencies of our circuits. The LPTC model
(see Lobula Plate Tangential Cell (LPTC) modelling section) has a decay time constant of
40 ms. The STMD model employs a temporal band-pass filter (early visual processing section
in S1 Appendix) for contrast adaptation, a temporal low-pass filter for the one-point corre-
lator, and an HR-detector for directional selectivity (see STMD modelling section in Meth-
ods), all together introducing delays of $\sim$ 16 ms. We quantified the resulting TSDN latency
– which we define as the time between the onset of a stimulus and a TSDN response $\geq$ 1 Hz
being elicited – after varying the target height (Fig 9A) or velocity (Fig 9B), for target trajec-
tories starting within the receptive field. No published hoverfly TSDN data is available so, for
biological comparison, we used data from dragonfly TSDNs, which have latencies between
20 ms and 40 ms [48]. For the relevant target height (1.4°) and velocity (90°/s), all three cir-
cuits exhibited latencies in this range (shaded areas, Fig 9). Assuming that hoverfly TSDNs
have similar latencies to dragonflies, we therefore assume that our TSDN circuits all have
comparable latencies to their biological counterparts.

We next investigated how the preceding optic flow configurations used by Nicholas
and Nordström [11] affect our models. They found that 1 s of preceding, syn-directional
optic flow inhibits biological TSDN responses to targets moving over a concurrent, sta-
tionary background, whereas preceding contra-directional optic flow does not facilitate
responses. An example stimulus consisting of such contra-directional preceding (right-
ward) optic flow with a leftward-moving target is presented in S1 Video. In our simulations,
the preceding optic flow lasted 0.48 s and was either stationary, syn-directional, or contra-
directional to target motion Fig 10. Importantly, as long as the optic flow is presented for at
least 80 ms, it stimulates LPTCs maximally (Fig 2A). Fig 10 shows the average TSDN response
throughout the 0.48 s period of target motion, averaged across neurons and conditions,
and normalised against the average TSDN response to a target moving 'alone' (green data
replotted from [11]). The results show that circuit 1 reproduces experimental results in all
three conditions (red, Fig 10). However, circuit 2 produces significantly different results to

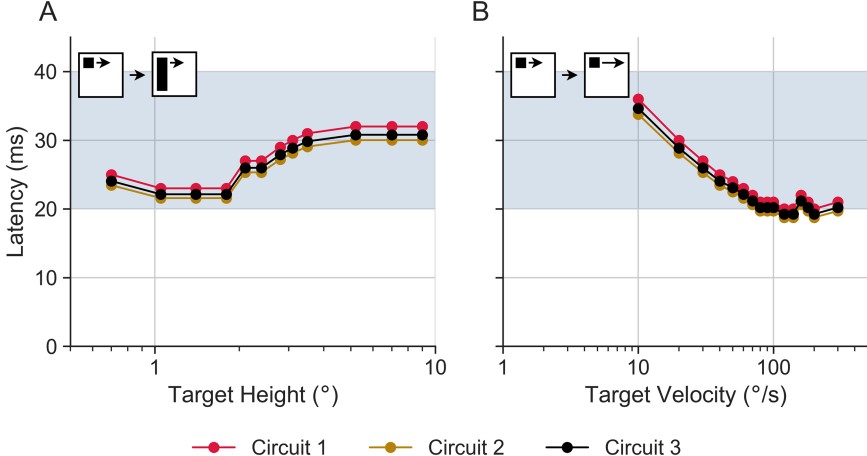

**Fig 9. Our TSDN circuits exhibit similar latencies to biological TSDNs.** TSDN model latencies (red, circuit 1;
golden, circuit 2; black, circuit 3) to changing spatiotemporal target characteristics. (**A**) Circuit latencies for different
target heights (fixed width of 0.8°, heights 0.7–9°, moving at 50°/s). (**B**) Circuit latencies for different velocities of
square targets of fixed size 0.8° × 0.8°. The velocity range is from 10–300°/s. The shaded region (20–40 ms) signifies
the latency range observed in dragonfly TSDNs [48]. Note that the pictograms are not to scale.

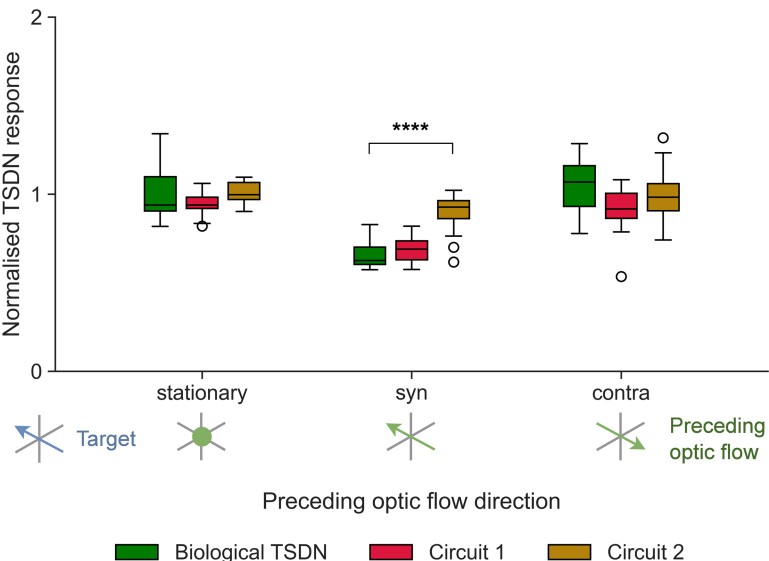

**Fig 10. Model and biological TSDN outputs to preceding optic flow.** The outputs of the fitted circuits are compared with biological TSDN responses (green, replotted from [11]). In the legend, target motion is shown in blue, and preceding optic flow with green arrows, or a dot, if stationary. The box edges denote the first and third quartiles of the data, with a line at the median. The whiskers denote the furthest data point within 1.5x the interquartile range (IQR). The unfilled circles are outliers. Significance is shown with one-way ANOVA followed by two-sample $t$-tests with Bonferroni correction (****$P \leq 0.0001$).

biological TSDNs for syn-directional preceding optic flow (golden, Fig 10, one-way ANOVA, followed by a post-hoc two-sample $t$-test with Bonferroni correction, $P \leq 0.0001$), suggesting that circuit 1 is more biologically plausible.

To better understand why syn-directional preceding optic flow inhibits TSDN responses, we simulated three targets that were spatially offset from each other by the distance the targets would travel in 25 ms (Fig 11). When the target starts within the TSDN receptive field (cyan target, Fig 11A, B) the modelled STMD response rises rapidly (orange, Figs 11C). When the starting position of the target is spatially offset to be outside the receptive field (green and red targets, Fig 11B), the timing of the STMD output is temporally delayed accordingly (orange, *middle* and *right*, Fig 11C).

Importantly, as the timing of the preceding optic flow is the same in these three conditions, the LPTC outputs in both the preferred (LPTC$_{PD}$) and anti-preferred directions (LPTC$_{ND}$) are also the same (green and purple, Fig 11C). Thus, when preceding optic flow in the preferred direction stops, LPTC$_{PD}$ activity decays back to the spontaneous rate (green, Fig 11C), with a time course similar to the LPTC response onset (*left*, Fig 2). At the same time, the inhibited LPTC$_{ND}$ returns to the spontaneous rate with a similar time course (purple, Fig 11C). Thus, when considering the model output in the 0.48 s analysis window (dashed box, Fig 11A), if the target starts within the receptive field (cyan target, Fig 11B), STMD activity coincides with decaying LPTC$_{PD}$ activity, and therefore the resultant TSDN output is suppressed ($\Delta t = 0$ ms, Fig 11C). However, if the targets start outside the receptive field, the LPTC activity has already returned to the spontaneous rate by the time the STMD activity starts, leaving the TSDN output unaffected ($\Delta t = 25$ ms and $\Delta t = 50$ ms, Fig 11C).

We find that in the model, TSDN firing rates averaged over the 0.48 s analysis window ($\bar{f}$) differ by 28.2% between the example where the target started well *outside* the receptive

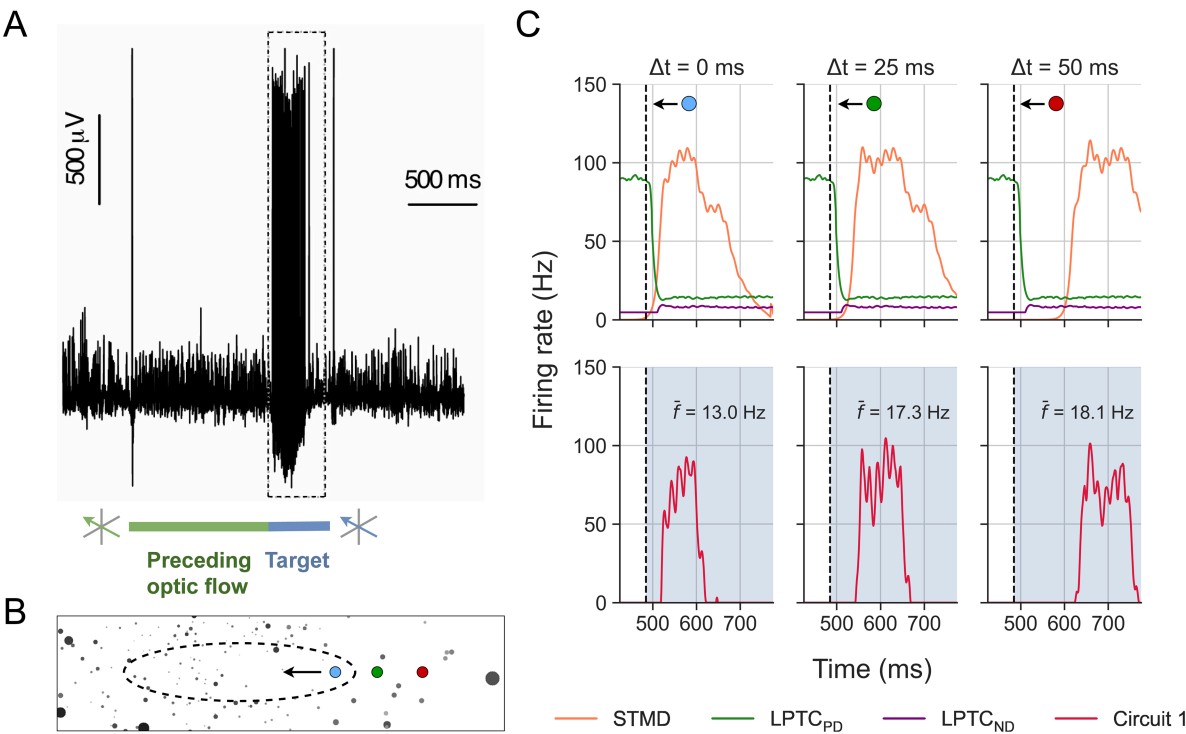

**Fig 11. The inhibition by preceding, syn-directional optic flow depends on target timing.** (A) Raw data trace from a TSDN to a target moving in its preferred direction. For 1 s before target motion, the 'starfield' background was sideslipping at 50 cm/s, in the TSDN neuron's preferred direction. During target motion, the starfield background was stationary. The coloured bars indicate the occurrence of the preceding optic flow (green) and the presence and movement of a target (blue). The dashed box indicates the (0.48 s) analysis window, which coincides with target motion. The trace is replotted from [11]. (B) Three modelled targets spatially offset from each other by the distance they would travel in 25 ms, on their trajectory into the modelled TSDN's receptive field (dashed ellipse). (C) *Top row* Modelled spike histograms as a function of time; STMD (orange), LPTC in preferred (green, LPTC$_{PD}$) and anti-preferred directions (purple, LPTC$_{ND}$). *Bottom row* Circuit 1 output for the three different target start positions as described in panel (B). The dashed lines indicate when the preceding optic flow stops and the target appears and starts moving. The shaded areas indicate the start of the 0.48 s analysis window.

field ($\Delta t$ = 50 ms, Fig 11C, 18.1 Hz) compared with when it started *within* the receptive field ($\Delta t$ = 0 ms, Fig 11C, 13.0 Hz). This response reduction is similar in magnitude to the difference observed in the biological TSDN firing rates between preceding optic flow that was either 'stationary' or 'syn-directional' to target motion (33.4%, green data, Fig 10). If the target starts at an intermediary position (green target, Fig 11B) the resulting firing rate in this example was 17.3 Hz (*middle*, Fig 11C).

Thus, we can now hypothesise why circuit 2 suppresses the TSDN response when syn-directional optic flow is concurrent with target motion (golden, Fig 5), but not when there is no concurrent optic flow, only preceding optic flow (golden, Fig 10). As circuit 2 receives input from two LPTCs, with opposite preferred directions of motion (circuit equations, Fig 4), these opposing LPTC activities cancel each other out in the preceding optic flow experiments (Fig 10). In contrast, when background optic flow is concurrent with target motion (see e.g. 'syn' in Fig 5), LPTC input in circuit 2 adequately suppresses the TSDN response, providing a good fit between the model and biology.

### Circuit 1 provides best fit to results of dot density experiments

In the experiments above, the density of the starfield background was kept at 100 dots/m$^3$. However, Nicholas and Nordström [11] also explored whether varying the dot density from 10–500 dots/m$^3$ affected the background modulation of the TSDN response. This is an important test as sparse widefield patterns generate weaker hoverfly LPTC responses than dense, high-contrast patterns [49], and we would therefore expect the LPTC input to be weaker at lower dot densities. Indeed, while optic-flow sensitive descending neurons respond strongly to the lowest dot density, the response is less than half of the response to the higher dot densities [11]. Our LPTC model output follows the same logarithmic behaviour as the dot density increases. Thus, the level of inhibition and facilitation should scale with dot density. However, while biological TSDN responses were not facilitated when low-density background motion was contra-directional to target motion, responses were facilitated at higher dot densities [11]. In contrast, biological TSDN responses were suppressed by syn-directional background motion at all dot densities tested. An example stimulus of the starfield background with a dot density of 500 dots/m$^3$ moving contra-directional to a leftward-moving target is presented in S2 Video.

Fig 12 shows the normalised (to each neuron's own mean response to a target moving over a white background) model and biological TSDN responses to a target moving against stationary, syn- and contra-directional background motion, across various dot densities. Both circuits 1 and 2 reproduce the suppression observed in biology (middle row, Fig 12). Circuits 1 and 2 sometimes exhibit suppression significantly stronger than biology; these happen because while the model outputs are completely driven to zero, in biology, TSDNs may experience noise that causes occasional firing (middle row, Fig 12)

However, when it comes to facilitation (which, in biology, depends on dot density [11]), circuit 2 sometimes produces TSDN outputs that are significantly different (asterisks in Fig 12). Circuit 2 performs worse because the amplification of the STMD activity by the LPTC with the opposite preferred direction (purple arrow, *middle* circuit, Fig 4) results in model outputs that *overshoot* the biological TSDN responses (see e.g. 'contra' panel, Fig 5). Thus, for contra-directional background motion at 10 dots/m$^3$, while circuit 2 outputs show facilitation, circuit 1 outputs match biological TSDN responses better (bottom row, Fig 12).

As modelled and biological LPTCs follow a logarithmic behaviour as dot density increases, our model suggests that, at a dot density of 10 dots/m$^3$ where the LPTC activity is low, there will be low/no facilitation of TSDN responses by contra-directional background motion. However, since even at a low dot density, syn-directional motion causes suppression of TSDN responses (middle row of Fig 12; [11]), this would indicate that the LPTC synaptic weight would have to be quite high relative to the STMD synaptic weight to fully suppress the STMD activity. This is confirmed in the LPTC synaptic weight ($\beta$) hyperparameter values obtained for both circuits 1 and 2 (shown in Table 1). Taken together, these last two experiments (Figs 10 and 12) again suggest that circuit 1 is the most biologically plausible.

## Discussion

We modelled three candidate TSDN circuits (Fig 4) using plausible combinations of existing STMD [29] and LPTC [20] models to drive the suppression by syn-directional background motion and the facilitation by contra-directional background motion seen in biology (Fig 7). We fitted our circuit models to published data [11], and then used new electrophysiology data to rule out one of the circuits (Fig 7; Fig 8). Then, we compared the remaining circuits against published preceding optic flow (Fig 10) and dot density (Fig 12) data [11]. Through the interplay of STMD and LPTC dynamics, our best-fitting model (circuit 1) provides mechanistic

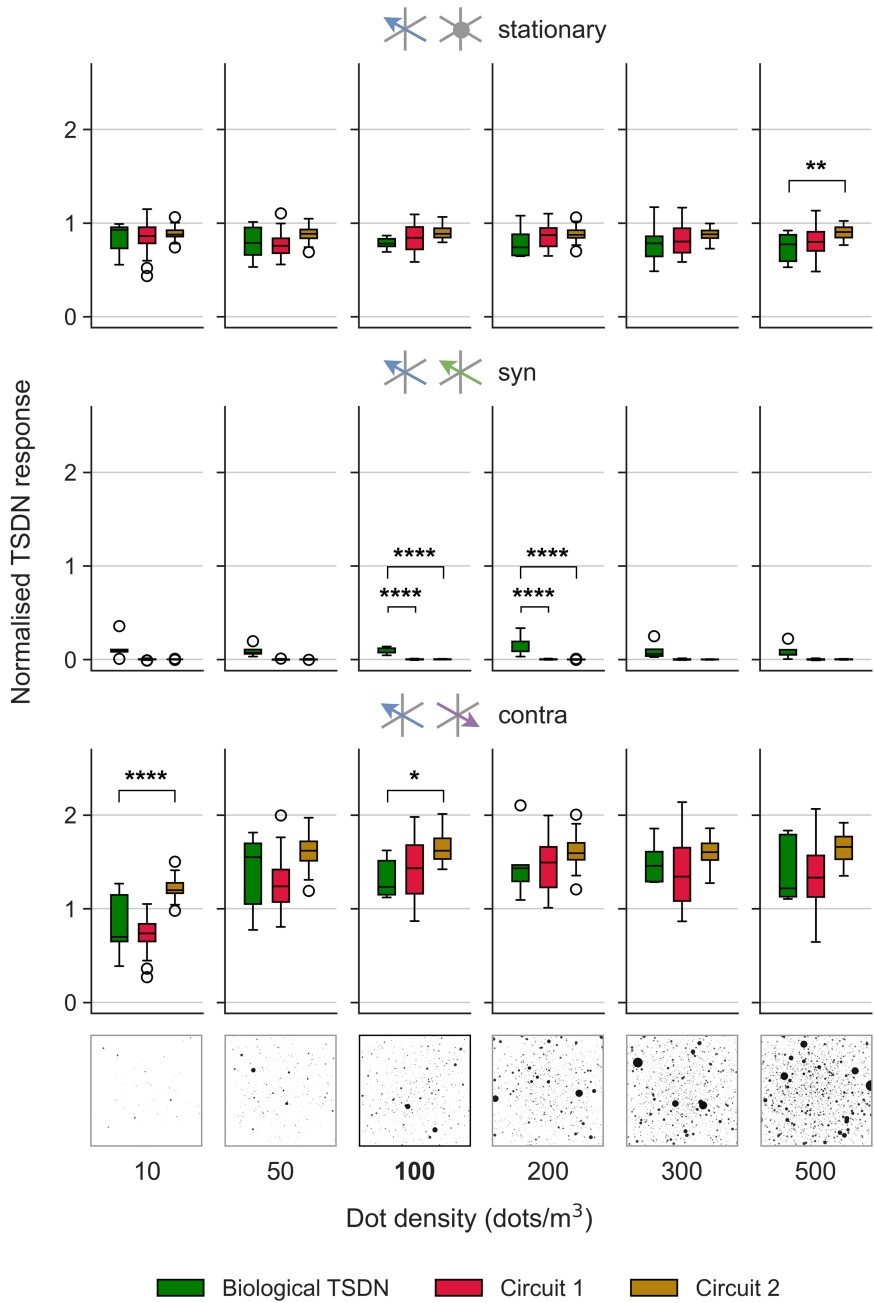

**Fig 12. Model and biological TSDN responses to targets moving over backgrounds with different dot densities.** The outputs of the fitted circuits (red, circuit 1; golden, circuit 2) and biological TSDN data (green data replotted from ref [11]) to targets moving over the starfield background with variable dot density. The different motion directions are colour-coded and results are laid out as such: target (blue), stationary (grey; top row), syn- (green; middle row), and contra-directional (purple; bottom row). The default starfield background at 100 dots/m³ is highlighted in bold. In each grid, significance is shown with one-way ANOVA followed by two-sample $t$-test with Bonferroni correction (*$P \leq 0.05$, **$P \leq 0.01$, ***$P \leq 0.001$, ****$P \leq 0.0001$).

explanations for suppression and facilitation of TSDN responses in the presence of syn- and contra-directional background motion (Figs 5 and 8), respectively, as well as observations related to preceding optic flow (Fig 10) and dot density (Fig 12) experiments [11].

Interestingly, the model that best fits the biological data (circuit 1, Fig 4) is the simplest one. Indeed, a similar circuit was proposed earlier [47] based on the finding that TSDN inhibition by background motion is proportional to the response amplitude of optic-flow sensitive neurons generated by that particular background. However, subsequent findings revealed asymmetries in the TSDN response [11]: facilitation from contra-directional background motion and inhibition from syn-directional motion were not balanced. These asymmetries, evident in the preceding optic flow (Fig 10) and dot density experiments (Fig 12), prompted the hypothesis that more complex mechanisms – such as those implemented in circuits 2 and 3 (Fig 4) – may be involved [11]. However, our modelling here shows that even the simple circuit can explain these findings (Figs 10 and 12).

Previous models have incorporated ESTMD and EMD output interactions [32,33]. However, they were either aimed at making ESTMD output directionally selective using an EMD mechanism [32] – a technique we also employ in making our directionally-selective STMD (see Directional-selectivity section in Methods) – or *improving* target detection in visual clutter [33], and thus were unable to capture the TSDN responses recorded in hoverflies [11,47]. In contrast, our modelling presented here not only explains the inhibition by syn-directional background motion but also the facilitation by contra-directional background motion seen in biology.

While there are more recent models of insect motion detection, such as the T4 model [24], we used a Hassenstein-Reichardt [20] detector to model LPTC responses. The main difference between the more recent T4 model and the EMD that *could* affect our results is the spatially offset anti-preferred direction responses in the T4 model at low background velocities. However, at the higher background velocity (90°/s) used here, responses to preferred and anti-preferred background motion follow a similar time course [24]. As such, we would not expect our findings to change dramatically with a different LPTC model, especially considering that our implementation fits biological data well with respect to temporal frequency tuning (Fig 1B), response onsets (Fig 2A), and preferred vs anti-preferred direction asymmetries (Fig 1).

While our deterministic model predicts how an *average* TSDN might respond to different target and background combinations, it cannot explain the variation between trials or between neurons seen in biology (e.g., Fig 7). In dragonflies, the equivalent neurons to TSDNs (i.e., those which specifically respond to small targets) vary vastly in receptive field size and location, direction selectivity, and response specificity [48], which could explain a large amount of inter-neuron variation. However, hoverfly TSDNs are more homogeneous with similar receptive fields and tuning characteristics [1], so heterogeneity is unlikely to explain the inter-neuron variability seen in our recordings (Fig 7). Therefore, the variability we see must come from other sources. One possible source could be the varying strengths of connections between neurons [50], which could lead to inter-neuron variability in TSDN responses. Furthermore, like dragonflies, hoverflies are poikilothermic animals with limited thermoregulation. Temperature has been shown to have a significant impact on the tuning properties of dragonfly STMD neurons [51], suggesting that any variations in temperature during recording sessions could lead to inter-trial variability. Finally, in blowflies, the response characteristics of LPTCs have been shown to depend on the activity state of the animal, i.e., LPTCs have different response characteristics depending on whether the animal is stationary, walking or flying [52]. As this state dependence can be mimicked by the application of chlordimeform (CDM, an octopamine receptor agonist), which has also been shown to affect hoverfly LPTCs

[53], it is plausible that activity- or attention-based LPTC modulation could contribute to inter-trial variability.

TSDNs are thought to contribute to target pursuit by supplying input to motor neurons [6]. Indeed, in dragonflies, TSDNs project to motor neurons responsible for wing control [48]. Additionally, other research [54] suggests that TSDNs may initiate foveating head movements, helping to stabilise the target image on the retina both before and likely during the 'shadowing' flight. Although it remains unclear whether hoverfly TSDNs directly influence head or wing movements, the simple circuit identified as the most parsimonious in our study (Fig 4) involves only minimal computational steps, suggesting it could support fast and accurate pursuit decision-making. Notably, if the hoverfly is self-generating optic flow, e.g., from body saccades, syn-directional background motion could result from a moving pursuer observing a stationary target, allowing the hoverfly to discriminate between *genuine* targets and target-like features (figure discrimination [31]), whereas contra-directional background motion may arise when the target is moving in the same direction as the pursuer. By comparing these signals, the pursuer could extract an error signal indicating the necessary corrective action. This idea is supported by research on predatory Diptera, which shows that they can use such error signals to guide efficient pursuit [55–57].

Interestingly, in response to target images reconstructed from pursuits [5], where the target is most often placed in the frontal part of the eye with binocular overlap [58], hoverfly TSDNs do not respond strongly [1]. However, these reconstructions were not completely natural as the hoverflies were pursuing artificial targets in a free-flight arena of limited size [5]. In nature, male *Eristalis* hoverflies set up territories, which they defend vigorously against other insects [59]. As they typically initially detect these intruding insects from a stationary stance, this would resemble our 'stationary' conditions. However, during pursuit, they might perform body saccades to keep the target in their visual field, and in this case, the background patterns would resemble our 'syn' and 'contra' conditions. In the future, it would be interesting to reconstruct both target images and optic flow generated during such target pursuit to determine how often the background moves 'syn'- and 'contra'-directional to the target. Indeed, it is likely that comparing the two signals is the key information needed for flight correction. Such an analysis would help evaluate the ethological impact of our findings.

## Materials and methods

### STMD modelling

Our STMD model implementation (Fig 13) can be partitioned into four stages: (1) 'Early visual processing', which includes the temporal and spatial filtering found in fly optics, photoreceptors and lamina monopolar cells; (2) 'Target matched filtering' which includes the processes that lead to the discrimination of the target against visual clutter; (3) 'Directional selectivity' which makes the resultant ESTMD directionally selective; (4) 'Receptive fields' to match the modelled STMD to that of a biological small-field STMD [34].

For stages (1-2), we reproduced the ESTMD model presented by Wiederman et al. [29] and Bagheri et al. [35]. We here added stages (3-4), providing directional selectivity (in a similar way to [32]) and small-field STMD receptive fields. We also tuned our ESTMD model hyperparameters to match hoverfly physiology. Details about the ESTMD model implementation are in S1 Appendix, whereas mechanisms introduced here are our contributions to extending the standard ESTMD model [29] to obtain a biologically plausible directionally selective small-field STMD model.

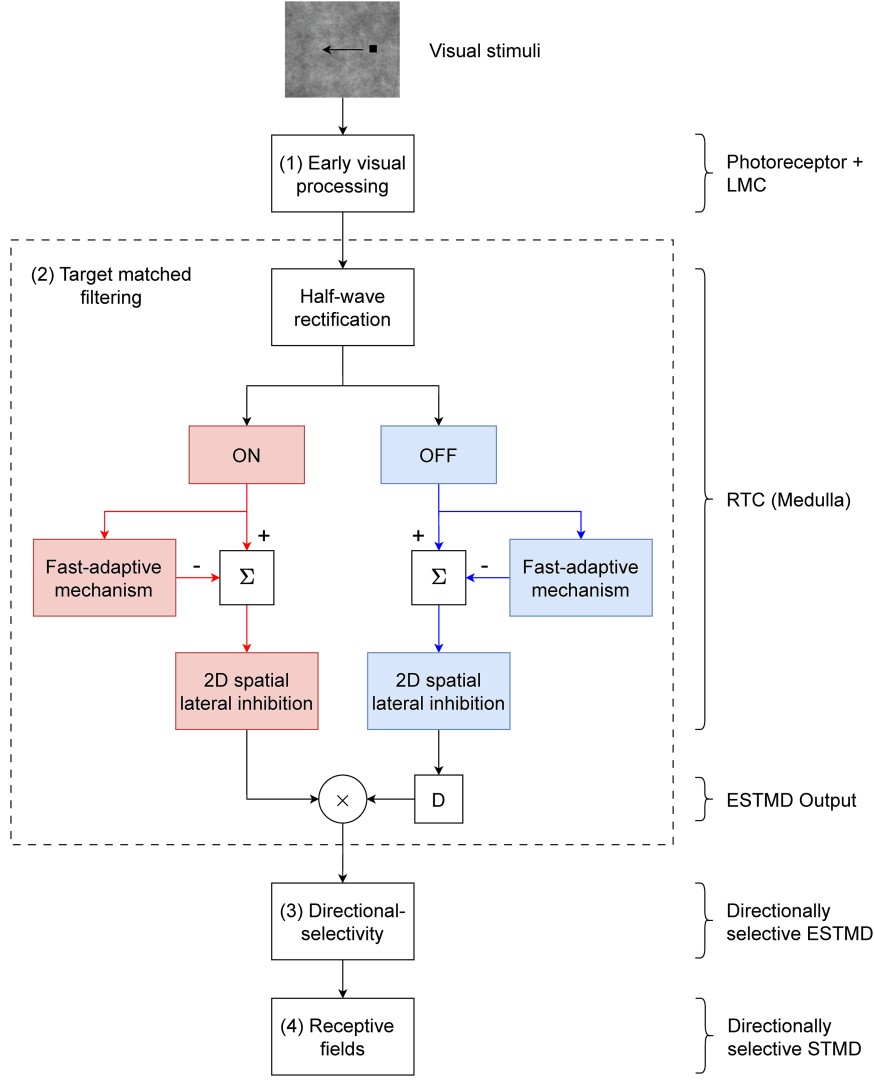

**Fig 13. STMD model.** Our STMD implementation – based on Wiederman et al. [29] with our additions to create directional selectivity (in a similar way to [32]) and mimic STMD receptive fields. Visual stimuli first undergo early visual processing in the fly photoreceptors and lamina monopolar cells (LMC). The LMC output is then separated into ON (red) and OFF (blue) channels, which are independently processed through rectifying transient cells (RTCs) found in the fly medulla. Then, the ESTMD output is a single-point correlation of a delayed OFF signal with an undelayed ON signal, as in [29,35]. The ESTMD output is then made directionally selective and spatially pooled using a 2D Gaussian representative of a small-field STMD's receptive field [34].

## Directional-selectivity

Since TSDNs are directionally selective [1], even in the absence of widefield motion, and based on our hypothesis that STMDs primarily drive the target selectivity seen in TSDNs, input from directional model STMDs seems plausible. To implement this, the non-directionally-selective ESTMD outputs are passed through a Hassenstein-Reichardt Elementary Motion Detector (HR-EMD) [17] (similarly to [32]), resulting in a preferred and an anti-preferred direction of target motion. As illustrated in Fig 14A, in the HR-EMD mechanism, two spatially segregated inputs ($x_1$ and $x_2$), separated by distance $\phi$, are correlated ($\times$)

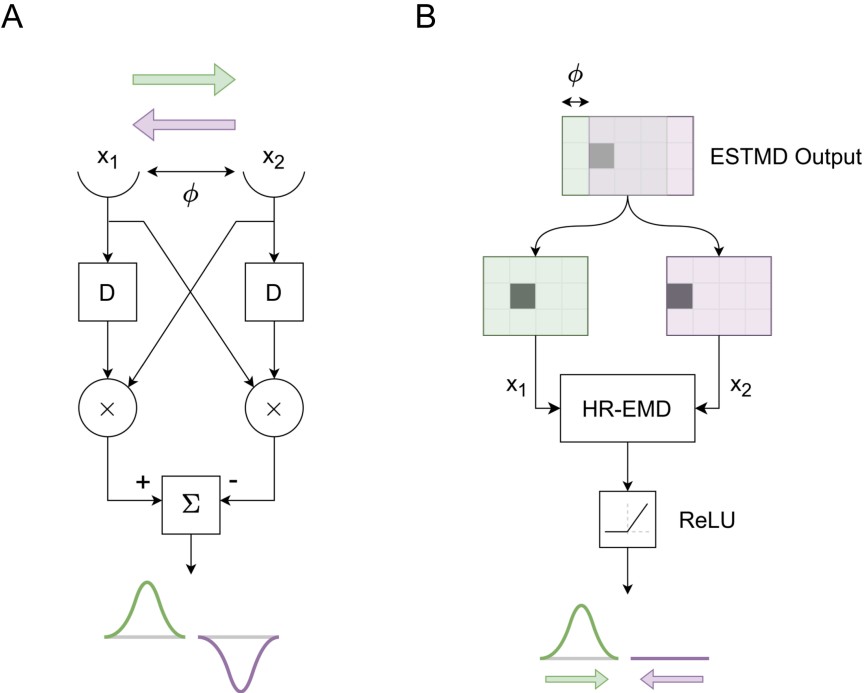

**Fig 14. HR-EMD mechanism leads to directional ESTMD output.** (**A**) The Hassenstein Reichardt EMD mechanism detects directional motion and produces a positive or negative response, denoted by a Gaussian, corresponding to the preferred (here rightward, green) or anti-preferred (here leftward, purple) direction. (**B**) The ESTMD output, represented as a 2D matrix, is split into two spatially segregated channels (green and purple matrices) that feed into the HR-EMD mechanism described in (A). The output is then half-wave rectified. In this case, the preferred direction of target motion is rightwards (green).

after delaying (D) one of the signals. The delay operator, D, like the one used in the Target matched filtering section in S1 Appendix, is a first-order low-pass filter with a time constant $\tau_D = 40$ ms. The correlation outputs are then subtracted, resulting in an HR-EMD output that gives positive output to one direction of motion, and negative output to the other.

The ESTMD output from 'target matched filtering' can be represented as a 2D matrix (*Top*, Fig 14B). Spatially segregating this matrix by a distance $\phi$, the spatially segregated inputs to the HR-EMD can be obtained. In this case, the green and purple shaded regions become $x_1$ and $x_2$ respectively (as described in Fig 14A). Finally, the HR-EMD output is passed through a Rectified Linear Unit (ReLU) block so that the output is non-negative.

## TSDN receptive fields

The final STMD output is obtained by spatially pooling directionally-selective ESTMD outputs (stage 4 in Fig 13). Barnett et al. [34] found that hoverfly small-field STMD receptive fields can be roughly expressed as a 2D isotropic Gaussian with $\sigma = 3°$. Thus, to achieve a good match with hoverfly biology, each STMD pools its inputs from a 2D Gaussian with $\sigma = 3°$. The receptive fields of multiple STMDs then tile the visual field with their centres separated by $2\sigma$ [60,61], resulting in a small overlap (Fig 15A).

The receptive field mapping method for biological TSDNs described in Nicholas and Nordström [11] leads to a coarse TSDN receptive field, shown with a green outline in Fig 15B. To improve the spatial resolution of the TSDN receptive field to a degree suitable for th

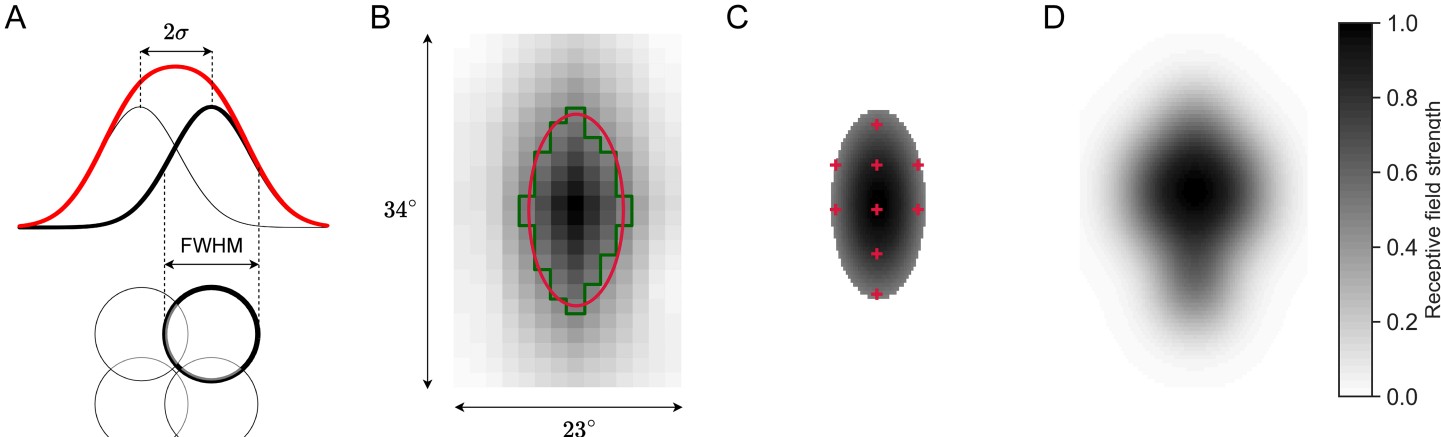

**Fig 15. Constructing STMD receptive fields.** (**A**) *Top*. The small-field STMD receptive field [34] is modelled as a Gaussian, shown in black. The distance between the centres of two adjacent receptive fields is then $2\sigma$, where $\sigma$ is the standard deviation of the Gaussian. Due to the Gaussians overlapping, the effective STMD receptive field strength is obtained through superposing (linear summating) the Gaussians. The superposed STMD receptive field is shown in red. *Below*. The STMD receptive fields at 50% maximum (full width at half maximum; FWHM), shown as circles, are arranged in a square grid. One model STMD receptive field and associated Gaussian are highlighted in bold. (**B**) A $34° \times 23°$ cropped patch of the visual field highlighting an example biological TSDN receptive field. The contours at 50% receptive field strength of the mapped TSDN receptive field (green) and a fitted 2D Gaussian (red). (**C**) The centres of the STMD receptive fields (red plusses) needed to create the biological TSDN receptive field in panel B are here overlaid on the 2D Gaussian. (**D**) The resulting modelled TSDN receptive field reconstructed from the superposed STMD receptive fields shown in panel C.

modelling, we fitted 2D Gaussians to the coarse biological TSDN receptive fields (red outline, Fig 15B). Since we presume that STMDs are presynaptic to TSDNs, we reconstructed the TSDN receptive field from model STMD receptive fields to ensure the model is TSDN receptive field-size invariant. In this way, since the receptive fields of small-field STMDs are smaller than those of TSDNs, multiple STMD outputs are pooled to give a single TSDN output. To calculate the number and locations of STMDs comprising each TSDN, we correlated the tiled STMDs across the visual field with the location of the TSDN receptive field (using the 50% Gaussian). In Fig 15C, the resultant STMD centres are shown as red plusses. Thus, by superposing the outstanding model STMD receptive fields, a reconstructed model TSDN receptive field was achieved (Fig 15D).

The widefield STMD used in circuit 3 (Fig 4) was modelled in a similar way to the small-field STMD. However, the large-field STMD spans over an azimuthal range of 0–30° and an elevation from 30–69° (the upper bound of the visual field stimulated in electrophysiology) ipsilateral to the TSDN receptive field [10].

## Lobula Plate Tangential Cell (LPTC) modelling

Optic-flow sensitive neurons in the fly optic lobe are known as Lobula Plate Tangential Cells (LPTCs). LPTCs integrate local motion signals from small-field T4 and T5 neurons [62]. In the absence of motion, spiking LPTCs fire at a spontaneous (baseline) rate and their activity increases or decreases from this baseline, depending on whether widefield motion is in the preferred direction (PD) or null direction (ND) (green data, Fig 1). However, the change in LPTC activity in response to widefield motion is asymmetric and dependent on the direction of the motion [63], i.e., the increase in LPTC activity is proportionally higher than the decrease in LPTC activity for the same amount of widefield motion in the preferred and null directions, respectively (green data, Fig 1).

We used the classic Hassenstein-Reichardt detector [20,22,23,64] model for our EMDs, and passed each ON-ON and OFF-OFF combination of inputs through a 2-quadrant-detector [65], to obtain two EMD outputs – one that has a PD to the right and another to the left (green and purple respectively in Fig 16). Each EMD output is half-wave rectified and spatially pooled. Spatial pooling creates a uniform directionally selective LPTC receptive field spanning across all elevations to either side of the azimuthal midline, e.g. left side of the visual field has LPTC with preferred direction to the left. The resultant ND response is multiplied by a ratio, $\gamma$, before subtracting from the resultant PD response (Fig 16) to provide an LPTC output. This linear subtraction imbues the LPTC output with an asymmetric sensitivity to widefield motion.

$\gamma$ (Fig 16A) is defined as the ratio of maximum increment to maximum decrement of the LPTC response about the spontaneous rate. The functional equation of the LPTC model involves the linear subtracted output (encompassed in curly brackets, Fig 16B) being scaled by a scaling coefficient, $g$, before being added to a non-zero spontaneous rate, $s$. The LPTC model response is clamped at zero as the spiking frequency cannot be negative. The delay operator in

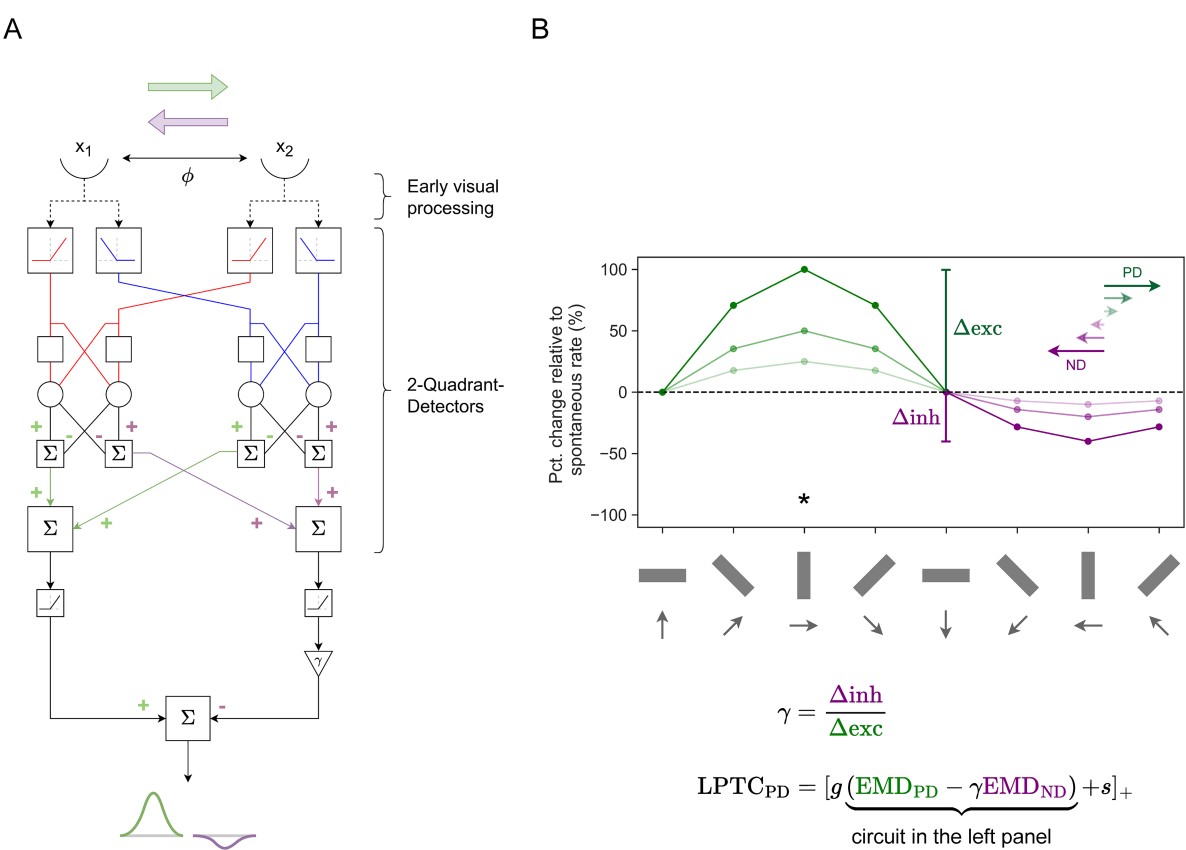

**Fig 16. LPTC modelling (A)** The inputs from two spatially-segregated points ($x_1$ and $x_2$) first undergo the same 'early visual processing' as the STMD model (Fig 13). Using 2-quadrant-detectors where only input combinations of the same contrast polarity are processed (ON-ON, OFF-OFF), directionally selective EMDs in the preferred direction (PD, green) and null direction (ND, purple) are obtained. In this case, the PD is to the right and the ND is to the left. The structural motifs in this circuit are the same as those in Fig 14A. (**B**) The direction sensitivity and the effect of different velocities of motion (longer and darker arrows denote higher velocity) in the PD and the ND on the modelled response. The grey arrows denote the direction of motion of the bar. In this case, the PD of motion is to the right, denoted by the *.

the 2-quadrant-detectors (blank square structural motif, Fig 16A), D, is expressed the same as in the Target matched filtering section in S1 Appendix, with a time constant, $\tau_D$ ($\tau_D = 40$ ms).

## Electrophysiology

Male *Eristalis tenax* hoverflies were immobilised at room temperature, ventral side up, using a mixture of beeswax and resin. A small region of the cuticle was removed at the anterior end of the thorax to expose the cervical connective. A fine wire hook was placed under the cervical connective for mechanical support, and a silver wire was inserted into the cavity as a reference electrode.

Extracellular recordings were made from the cervical connective using a sharp polyimide-insulated tungsten microelectrode (2 MΩ; Microprobes). Data were amplified at 100 gain and filtered through a 10–3000 Hz bandwidth filter using a DAM50 differential amplifier (World Precision Instruments), with 50 Hz noise removed with a Hum Bug (Quest Scientific). Data were acquired and digitised at 40 kHz using Powerlab 4/30 and LabChart 7 Pro (ADInstruments). Single units were discriminated by amplitude and half-width using Spike Histogram software (ADInstruments).

During the experiment, hoverflies were positioned perpendicular to and 6.5 cm away from the middle of a linearised LCD screen (ASUS) with a mean illuminance of 200 lx, a refresh rate of 165 Hz, and a spatial resolution of 2560×1440 pixels (59.5×33.5 cm), giving a projected screen size of 155° × 138°. Visual stimuli were displayed using custom software written [66] in MATLAB (version R2019b; MathWorks) using the Psychophysics toolbox [67,68].

TSDNs were identified as described [9]. In short, we mapped the receptive field of each neuron by scanning a target horizontally and vertically at 20 evenly spaced elevations and azimuths [1] to calculate the local motion sensitivity and local preferred direction. We then scanned targets of varying height through the small, dorso–frontal receptive fields to confirm that each neuron was sharply size tuned with a peak response to targets subtending 3° to 6°, with no response to larger bars, to looming or widefield stimuli [1,9].

## Visual stimuli

To test the effects of background motion on the TSDN response, we displayed targets against one of three different backgrounds (shown in Fig 17). Unless otherwise mentioned, targets were black and round with a diameter of 15 pixels (starfield background) or 15 × 15 pixels black squares (cloud and sinusoidal backgrounds), moving at a velocity of 900 pixels/s for 0.48 s. When converted to angular values and taking the small frontal TSDN receptive fields into account [1], this corresponds to an average diameter of 3° and a velocity of 130°/s [47]. Unless otherwise stated, each target travelled in each neuron's preferred horizontal direction (i.e., left or right) and across the centre of its receptive field. The target elevation was varied slightly between repetitions to minimise habituation [11]. There was a minimum of 4 s between each stimulus presentation.

The background stimuli all covered the entire visual monitor, i.e. they had a spatial extent of 2560 × 1440 pixels, corresponding to 155° × 138° [11]:

**Starfield**  a 2D rendering of a simulated 3D environment containing randomly located spheres. These were perspective corrected to display the optic flow that would be generated if the hoverfly was sideslipping at 50 cm/s, with size and greyscale indicating their relative distance from the hoverfly (see ref [14] for more details). Some of the spheres had the same spatio-temporal characteristics as the target.

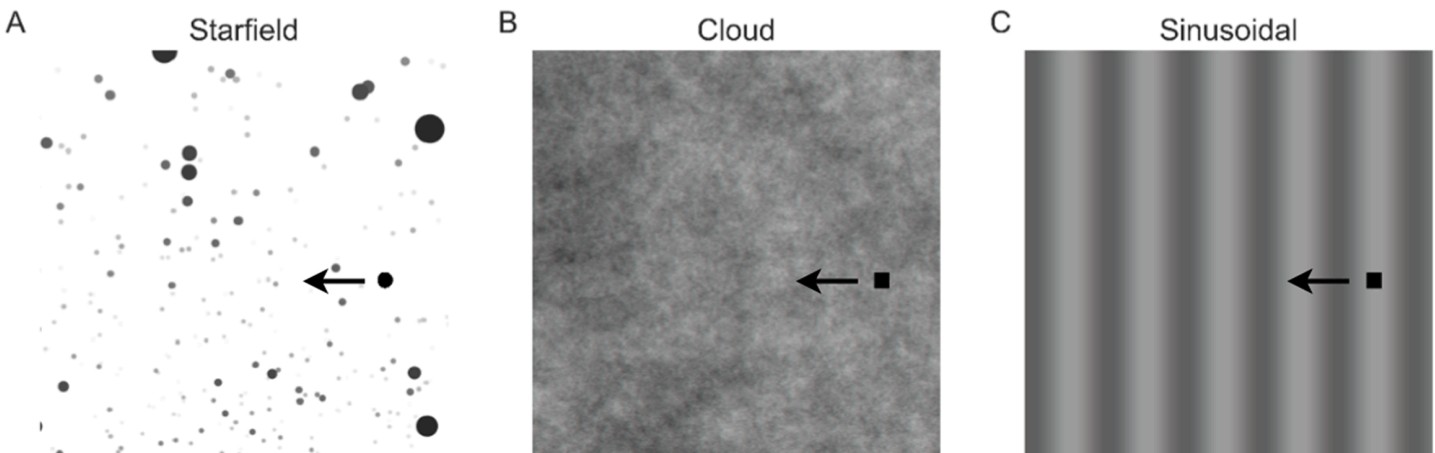

**Fig 17. Target-background combinations.** The three background stimuli used for both electrophysiology and simulations: (**A**) Starfield, (**B**) Cloud, (**C**) Sinusoidal. The target for the background in (**A**) was 15 pixels diameter black dots moving at 900 pixels/s. The targets for the backgrounds in (**B,C**) were 15 pixels side length black squares moving at 900 pixels/s. The targets traversed the background in the preferred direction of each TSDN – in this case, to the left. Note that the targets are not to scale.

**Cloud** a greyscale background with a 2D power spectrum similar to that of typical natural scenes and a mean luminance of $\sim 128$ where luminance $\in [0, 255]$ [47] with a Michelson contrast of 100% (see ref [7] for more details).

**Sinusoidal** a vertical sinusoidal grating with an average wavelength of 7°, drifting at 5 Hz, and a Michelson contrast of 25%.

For each background, four conditions were tested. (1) 'alone': the target moved across a homogeneous white (control for starfield) or grey (controls for cloud and sinusoidal) background; (2) 'stationary': the target moved over a non-moving background; (3) 'syn'-directional: the background moved in the same direction as the target; (4) 'contra'-directional: the background moved in the opposite direction to the target. The same stimuli shown to the hoverflies were also presented to the model as a sequence of images, each presented for 1 ms.

## Data analysis and statistics

We recorded from N=10 TSDNs (cloud) and N=6 TSDNs (sinusoidal) in 16 male hoverflies. We kept data from all TSDNs that showed a robust response to a target moving over a white background. We repeated this control throughout the recording and only kept data from neurons that responded consistently. We only kept data from experiments with a minimum of 12 repetitions. The data from repetitions within a neuron were converted to rates and averaged per condition. We normalised the responses to each neuron's own mean response to a target moving over a white or grey background.

To compare the TSDN spike trains with the output of our rate-based model, we convolved them with Gaussian kernels to obtain a Spike Density Function (SDF) [69]:

$$s(t) = \sum_i \kappa(t - t_i) \tag{1}$$

$$\kappa(t) = \frac{1}{\sigma\sqrt{2\pi}} \exp\left(-\frac{t^2}{2\sigma^2}\right) \tag{2}$$

where $t_i$ are the times of individual spike and $\sigma = 20$ ms is the standard deviation of the Gaussian kernel $\kappa$.

Data analysis of electrophysiology data was performed in MATLAB. Throughout the paper, n refers to the number of repetitions within one neuron and N to the number of neurons. The sample size, type of test, and *P* value are indicated in the corresponding figure legends. For the cloud and sinusoidal electrophysiological recordings (Fig 7), paired *t*-tests or one-way ANOVAs, followed by Dunnett's post hoc test for multiple comparisons, were used (using Prism 10.4.1 for Mac OS X (GraphPad Software)). For the preceding optic flow (Fig 10; S1 Video) and dot density results (Fig 12; S2 Video), one-way ANOVA followed by two-sample *t*-tests with Bonferroni correction for multiple comparisons was used.

## TSDN candidate circuit cost function

In the electrophysiology experiments, the target elevation was varied slightly between repetitions to minimise habituation (Section Visual stimuli, Nicholas and Nordström [11]). At each target elevation, a single time-series produced by our circuit was compared against the repetition-averaged time series of spike rates obtained from our electrophysiological recordings (see section Data analysis and statistics) to give a root-mean-square error (RMSE). The RMSEs from each dimension of the electrophysiology recordings were averaged, i.e., over elevations (*E*), conditions (*C*), e.g., 'alone' or 'stationary', and neurons (*N*). Thus, the total RMSE ($RMSE_{total}$) that a circuit yields is expressed as:

$$RMSE_{total} = \frac{1}{NCE} \sum_{n=0}^{N} \sum_{c=0}^{C} \sum_{e=0}^{E} \underbrace{\sqrt{\frac{1}{T} \sum_{t=0}^{T} (y_{t,e,c,n}^{circuit} - y_{t,e,c,n}^{ephys})^2}}_{\text{RMSE per target elevation}} \qquad (3)$$

where $y^{circuit}$ is the deterministic circuit output and $y^{ephys}$ is the repetition-averaged TSDN spike rates – both time-series. When fitting against the starfield background, the aim was to minimise the $RMSE_{total}$ to obtain the optimal circuit and LPTC coefficients ($\alpha$, $\beta$, $\kappa$, $\eta$, $g$, $\gamma$ and $s$) (Fig 16, Table 1). However, when testing the performance of the circuits against the cloud and sinusoidal backgrounds, we used the $RMSE_{total}$ for each circuit normalised by the $RMSE_{total}$ of the 'STMD only' circuit (Fig 5).

## Supporting information

**S1 Appendix. Implementation details of the ESTMD model.**
(PDF)

**S1 Video. Example preceding optic flow video.** Video at 60 fps of an example stimulus used in the preceding optic flow experiments. The preceding optic flow is contra-directional (rightward) while the concurrent optic flow is stationary. The target moves leftward. The black ellipse indicates the boundary of the particular TSDN's receptive field at 50% of the maximum response. Data from [11].
(MP4)

**S2 Video. Example dot density video.** Video at 60 fps of an example stimulus used in the dot density experiments. The starfield background has a dot density of 500 dots/m$^3$ and is moving contra-directional to the target (rightward). The target moves leftward. The black ellipse

indicates the boundary of the particular TSDN's receptive field at 50% of the maximum response. Data from [11].
(MP4)

## Acknowledgments

SN and KN thank Biomedical Engineering at SAHLN and the Botanic Gardens of Adelaide for their ongoing support.

## Author contributions

**Conceptualization:** Anindya Ghosh, Sarah Nicholas, Karin Nordström, Thomas Nowotny, James Knight.

**Data curation:** Anindya Ghosh, Sarah Nicholas.

**Formal analysis:** Anindya Ghosh, Sarah Nicholas.

**Funding acquisition:** Karin Nordström, Thomas Nowotny, James Knight.

**Investigation:** Anindya Ghosh, Sarah Nicholas.

**Methodology:** Anindya Ghosh, Sarah Nicholas, Karin Nordström, Thomas Nowotny, James Knight.

**Project administration:** Karin Nordström, Thomas Nowotny, James Knight.

**Resources:** Karin Nordström, Thomas Nowotny, James Knight.

**Software:** Anindya Ghosh.

**Supervision:** Karin Nordström, Thomas Nowotny, James Knight.

**Validation:** Anindya Ghosh, Sarah Nicholas.

**Visualization:** Anindya Ghosh, Karin Nordström.

**Writing – original draft:** Anindya Ghosh.

**Writing – review & editing:** Sarah Nicholas, Karin Nordström, Thomas Nowotny, James Knight.

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
