## [Decision Letter · Decision Letter 0]

30 Apr 2025

PCOMPBIOL-D-25-00548

Understanding the mechanism of facilitation in hoverfly TSDNs

PLOS Computational Biology

Dear Dr. Ghosh,

Thank you for submitting your manuscript to PLOS Computational Biology. After careful consideration, we feel that it has merit but does not fully meet PLOS Computational Biology's publication criteria as it currently stands. Therefore, we invite you to submit a revised version of the manuscript that addresses the points raised during the review process.

Please submit your revised manuscript within 60 days Jun 30 2025 11:59PM. If you will need more time than this to complete your revisions, please reply to this message or contact the journal office at ploscompbiol@plos.org. Please include the following items when submitting your revised manuscript:

We look forward to receiving your revised manuscript.

Kind regards,

Barbara Webb

Academic Editor

PLOS Computational Biology

Marieke van Vugt

Section Editor

PLOS Computational Biology

**Journal Requirements:**

At this stage, the following Authors/Authors require contributions: Anindya Ghosh, Sarah Nicholas, Karin Nordström, Thomas Nowotny, and James Knight. Please ensure that the full contributions of each author are acknowledged in the "Add/Edit/Remove Authors" section of our submission form.

5) We have noticed that you have uploaded Supporting Information files, but you have not included a list of legends. Please add a full list of legends for your Supporting Information files after the references list.

**Reviewers' comments:**

Reviewer's Responses to Questions

**Comments to the Authors:**

Reviewer #1: In this work, Ghosh et al develop computational models of TSDNs in the hoverfly, Eristalis tenax. They test different model variants to elucidate which best represents observed physiological results. The models are derived from interactions between small-field STMDs and wide-field, optic-flow neurons (LPTCs).

The manuscript is well written and clear (except for the section on preceding motion). The modelling itself is incremental in its complexity, given that there has been published models exploring interactions between LPTCs and ESTMDs. The parsimonious Circuit 1 would seem to be an obvious solution for the TSDN responses (including facilitation). However, since these interactions have been examined with respect to a different physiological system (TSDN in hoverfly), as well as presenting physiological results, makes this an impactful and novel contribution.

Comments

Line 18: supressed or facilitated. It would be nice to describe more completely here compared to what? Presumably a target on a blank screen?

Line 27: Models in references 16,17 used a matched temporal filter in addition to strong surround antagonism.

Line 40: Perhaps reference the ESTMD modelling that included directionality, implemented in a similar way.

S. D. Wiederman and D. C. O'Carroll, "Biomimetic target detection: Modeling 2nd order correlation of OFF and ON channels," 2013 IEEE Symposium on Computational Intelligence for Multimedia, Signal and Vision Processing (CIMSIVP), Singapore, 2013, pp. 16-21, doi: 10.1109/CIMSIVP.2013.6583842.

Line 48: Perhaps reference the ESTMD modelling that interacted with LPTC neurons in a similar way.

Steven D. Wiederman, Russell S. A. Brinkworth, David C. O'Carroll, "Bio-inspired target detection in natural scenes: optimal thresholds and ego-motion," Proc. SPIE 7035, Biosensing, 70350Z (2008); https://doi.org/10.1117/12.804351

Following this work, there were more labs that modelled such LPTC-ESTMD interactions (see several Brinkworth articles).

Line 77: Rather than blowfly, should this include citations with Eristalis, which has been an animal model for LPTC investigations (e.g. temporal frequency tuning)

Line 104: this makes for a dark bar detector. It is the inclusion of strong centre-surround antagonism which creates the target selectivity.

Line 120. At the same speed? Or doesn’t this matter?

Line 121: As above, the relative speed relationships should be described (if known).

Line 130: “opposite preferred direction” Nice to be clear whether describing the neuron’s directionality or the stimulus direction

Line 117: In this section, it is not clear whether the neurons exhibit direction opponency, that is, do they produce below spontaneous levels for motion in the non-preferred direction?

Line 136: Wouldn’t this be the same for Circuit 2?

Line 147: In hoverflies, LPTC receptive fields have been mapped with target-like objects, as they induce large responses (presumably non-linear spatial weighting). Therefore, it’s not very clear whether star-field is designed as an optic flow or target stimulus, which makes this section difficult to follow.

Line 154: fitted to spike rates of peristimulus time histograms?

Line 173: The point about spontaneous activity and whether the LPTC models are direction opponent should be made more explicit.

Line 175: What does a “more involved navigation” mean?

Figure 5: how does pixels/s relate to cm/s

Line 204: “could theoretically”. Does it or does it not stimulate STMDs? If so, what does this mean for the interpretation of results?

Liner 211: What is surprising? Is it a discrepancy between prior work? If so, then describe completely so that future research may resolve conflict in the literature.

Line 222: Is there a rationale for why Circuit 3 should be a possible architecture?

Line 234: overshooting characteristic is not clear

Line 236: if introduced earlier, the concept of target-like objects within forms of background motion would make more sense (rather than a description following)

Line 246: how does immediately precede relate to latencies? This entire section has limited logical flow between undefined physiological data. How long is the preceding stimulus? When are the TSDN responses analysed following? If the rationale is that decay offsets of LPTC neurons affect following TSDN responses, then introduce this concept first.

Line 250: If TSDN responses are suppressed by preceding optic flow, then doesn’t this change the model validation (i.e. that you are building correct models). Given this is the case, it is very important to provide clarity on these attributes, e.g. ‘preceding’ durations. Moreover, how latency and decay offsets are proposed to underlie model verification. Overall, how latencies of LPTCs and TSDNs relate to both physiological responses (of ‘preceding’ experiments) as well as model parameters (e.g. low pass filters) is not clear enough (similarly, Figure 10).

Figure 9: Do these bar sizes start within the receptive field? If not, how are latencies calculated for stimuli that drift into a neuron’s receptive field.

Line 316: The manuscript should explain the spatial weighting of LPTCs observed physiologically and in the modelling. That is, would a hoverfly LPTC respond more strongly to a dot density of 10 dot/m2. Would response be similar above 50 dots/m2?

Line 329: “plausible combinations” not clear why these architectures were chosen

Line 339: What metric determines whether these results are variable?

Line 379: If TSDNs drive foveation, then whether contra or syn directional target and background stimuli covers the visual field of single or both eyes could be very important.

Refs – some typo errors

Reviewer #2: Review: Understanding the mechanism of facilitation in hoverfly TSDNs

In this manuscript the authors test 3 different models that could explain the way in which Target Descending Selective Neurons (TSDNs) receive two types of movement inputs:

from widefield cells, which carry information about self motion

from small target neurons, which carry motion information about a potential prey/mate motion.

The manuscript is well laid out, and the figures are clear. I have a few queries.

1 As far as I can tell, the modelling for self-motion information is based on Reichardt Correlators, as published by Borst et al groups. But there seems to be no mention of more recent publications, for example from Michael Reisers group, showing that the fast excitatory and offset delayed inhibition is what produces the directional sensitivity in flies. This is fundamentally different from the Reichardt Correlator. See Figure 6 in Gruntman et al 2018. I can not understand how such reference is missing in a diptera paper about motion computation.

Could the authors use a t4 model instead of Reichardt Correlators? Would this alter the results of the model?

Even if the results are not altered, the fact that the Reichardt Correlators is a model now superseded by current evidence should be mentioned and clearly explained.

2. In the intro, there seems to be no mention about the real world importance of both types of information getting to the TSDNs. This is only mentioned at the very end of the conclusion, but I think that to ground the reader, it should be included in the introduction. This also sets the stage for explaining why the same motion for target and background results in minimal TSDN activity (the target didn’t move, the pursuer did), or why the background moving in the opposite direction to the target should facilitate the TSDN activity (the target is moving faster the pursuer…and it is therefore slipping away). the biological system needs to compare both signals (is the target motion detected at the retina level caused by self-motion, or is the target moving in a way that differs from the predator). Comparing the two signals gives the true target motion that needs to be corrected, and indeed is the signal necessary for a proportional navigation system which explains predation in other diptera. This computation needs to be fast and accurate: which would push the system to be robust and minimal number of computation steps. This would all argue for the 1st model to be the most likely one.

3. I must be missing something when it comes to the 1st part of the results: Model 3 is a complex model made to fit the very specific conditions in which the data was collected,. It is not surprising that it was the best performing model for that data set and that it does not fit anything else well. At that point in the paper, this set off alarm bells for the reader: Do the authors not understand that they have artificially created a fit? It is great that the authors went ahead and collected more data to show that this was indeed the case, but maybe an acknowledgement that Model 3 was expected to be the best performing because it design to fit the specific circumstances of that data set, would warn the readers that the authors were aware of these pitfalls.

4. the authors mention that TSDNs are downstream of the other target neurons. Do we KNOW that? Or is it believed that this is the case?

5. The authors mention the variability of the TSDN responses, but fail to mention that TSDNs are not a homogeneous population. Much of the paper is based on dragonfly data, but dragonfly TSDNS have clearly different roles, in addition to target selectivity, some of them are looming sensitive and others carry predictive information. What type of TSDNs did the authors record from? Do they know it was always the same type?

Line 15. We don’t know that they are postsynaptic partners right?

Line 76. The model fails to capture the width of the response, with the biological one being wider. This doesn’t seem addressed. Could the authors please explain the reason behind this discrepancy?

Lines 87-97 – the jump between those two paragraphs is difficult for a reader to follow. A rewrite may help.

Figure 3: here the biological curve is wider again. Why is that?

Line 168 ‘ the STMD output remains unchanged’. Is this model or biological data.? This isn’t made clear to the reader. Same for label of Figure 5 blue line.

Figure 7: D: there are more spikes for this treatment, but the DC line seems to go lower. Is there a sustained inhibition then? This seems to occur in other treatments but it is not addressed?

Figure 7: 3 different TSDNs. Which TSDNs? Did they have the same properties?

Line 366. This section appears to belong in Results, not discussion? Also, why are the results not shown? They should always be included somewhere.

Q on methods: As mentioned by the authors TSDNs are not rate neurons. The authors use a convolution method to turn the data into Spike Density Function. But they give no reference for this method. This is important for the reader that wants to know how appropriate it is…etc For example,is it a fixed kernel? Please provide a reference.

**Have the authors made all data and (if applicable) computational code underlying the findings in their manuscript fully available?**

Reviewer #1: Yes

Reviewer #2: Yes

PLOS authors have the option to publish the peer review history of their article (what does this mean?). If published, this will include your full peer review and any attached files.

Reviewer #1: No

Reviewer #2: No

**Figure resubmission:**
---

## [Decision Letter · Decision Letter 1]

26 Aug 2025

PCOMPBIOL-D-25-00548R1

Understanding the mechanism of facilitation in hoverfly TSDNs

PLOS Computational Biology

Dear Dr. Ghosh,

Thank you for submitting your manuscript to PLOS Computational Biology. After careful consideration, we feel that it has merit but does not fully meet PLOS Computational Biology's publication criteria as it currently stands. Therefore, we invite you to submit a revised version of the manuscript that addresses the points raised during the review process.

The attached review shows that one of the previous reviewers was satisfied with the changes. As the other reviewer was unavailable, as editor I have carefully followed the response to reviewers and read the manuscript and suggest the following minor points should be addressed:

1) It is helpful to have added mention and some discussion of the T4/T5 models but their introduction could be clearer: these are not just 'other recent' models but models far more strongly justified by biological evidence than the preceding phenomenological models. 

2) In the author summary, the problem is described as 'computationally hard' but nothing in the text explains why it is hard (indeed it seems a simple model is sufficient). Both the summary and abstract imply the solution here could be of relevance to bio-inspired robotics but this receives no further mention anywhere in the paper. Either add some relevant discussion about how the equivalent problem is currently addressed in robotics and/or computer vision, or drop these references to robotics.

Please submit your revised manuscript within 30 days Oct 26 2025 11:59PM. If you will need more time than this to complete your revisions, please reply to this message or contact the journal office at ploscompbiol@plos.org. Please include the following items when submitting your revised manuscript:

We look forward to receiving your revised manuscript.

Kind regards,

Barbara Webb

Academic Editor

PLOS Computational Biology

Marieke van Vugt

Section Editor

PLOS Computational Biology

**Reviewers' comments:**

Reviewer's Responses to Questions

**Comments to the Authors:**

Reviewer #1: I have read through the substantially revised manuscript and am satisfied that the authors have addressed comments in this new version. Overall, the manuscript has been improved by providing additional clarity in many sections.

**Have the authors made all data and (if applicable) computational code underlying the findings in their manuscript fully available?**

Reviewer #1: Yes

PLOS authors have the option to publish the peer review history of their article (what does this mean?). If published, this will include your full peer review and any attached files.

Reviewer #1: No

**Figure resubmission:**
---

## [Editor Report · Decision Letter 2]

30 Sep 2025

Dear Mr Ghosh,

We are pleased to inform you that your manuscript 'Understanding the mechanism of facilitation in hoverfly TSDNs' has been provisionally accepted for publication in PLOS Computational Biology.

Best regards,

Barbara Webb

Academic Editor

PLOS Computational Biology

Marieke van Vugt

Section Editor

PLOS Computational Biology

---

## [Editor Report · Acceptance letter]

PCOMPBIOL-D-25-00548R2

Understanding the mechanism of facilitation in hoverfly TSDNs

Dear Dr Ghosh,

I am pleased to inform you that your manuscript has been formally accepted for publication in PLOS Computational Biology. Your manuscript is now with our production department and you will be notified of the publication date in due course.

With kind regards,

Zsofia Freund
